# Active thrombin produced by the intestinal epithelium controls mucosal biofilms

Jean-Paul Motta [1,6], Alexandre Denadai-Souza[1,6], David Sagnat[1], Laura Guiraud[1], Anissa Edir[1], Chrystelle Bonnart[1], Mireille Sebbag[1], Perrine Rousset[1], Ariane Lapeyre[1], Carine Seguy[1], Noa Mathurine-Thomas[1], Heather J. Galipeau[2], Delphine Bonnet[3], Laurent Alric[3], Andre G. Buret[4], John L. Wallace[5], Antoine Dufour [5], Elena F. Verdu[2], Morley D. Hollenberg[5], Eric Oswald[1], Matteo Serino[1], Celine Deraison[1] & Nathalie Vergnolle [1,5]

Proteolytic homeostasis is important at mucosal surfaces, but its actors and their precise role in physiology are poorly understood. Here we report that healthy human and mouse colon epithelia are a major source of active thrombin. We show that mucosal thrombin is directly regulated by the presence of commensal microbiota. Specific inhibition of luminal thrombin activity causes macroscopic and microscopic damage as well as transcriptomic alterations of genes involved in host-microbiota interactions. Further, luminal thrombin inhibition impairs the spatial segregation of microbiota biofilms, allowing bacteria to invade the mucus layer and to translocate across the epithelium. Thrombin cleaves the biofilm matrix of reconstituted mucosa-associated human microbiota. Our results indicate that thrombin constrains biofilms at the intestinal mucosa. Further work is needed to test whether thrombin plays similar roles in other mucosal surfaces, given that lung, bladder and skin epithelia also express thrombin.

[1] IRSD, Université de Toulouse, INSERM, INRA, ENVT, UPS, U1220, CHU Purpan, CS60039, 31024 Toulouse, France. [2] Farncombe Family Digestive Health Research Institute, McMaster University, Health Science Center, Rm 3N4, 1280 Main Street West, Hamilton, ON L8S 4K1, Canada. [3] Department of Internal Medicine and Digestive Diseases, 1, avenue Jean Poulhes-TSA 50032, 31059 Toulouse, France. [4] Department of Biological Sciences, University of Calgary, 2500 University Drive NW, Calgary, Ab T2N 4N1, Canada. [5] Departments of Physiology & Pharmacology, and Medicine, University of Calgary Cumming School of Medicine, 3330 Hospital Drive NW, Calgary, Ab T2N 4N1, Canada. [6] These authors contributed equally: Jean-Paul Motta, Alexandre Denadai-Souza. Correspondence and requests for materials should be addressed to N.V. (email: nathalie.vergnolle@inserm.fr)

Epithelia can serve as a local source of proteases at the surface of mucosal organs such as in the skin[1,2], the lung[3] and the digestive tract[4,5]. Previous work indicates that bacterial infection[6], stress[5], ischemia[7], low-grade[8] or high-grade inflammation[9,10] can trigger mucosal and epithelial release of active proteases by the host. Functional proteomic profiling has recently identified active proteases released by human intestinal mucosa in health and in inflammatory bowel diseases[11]. Among the identified active proteases, thrombin was found to be present both in healthy and inflamed human mucosa, although the cellular source of thrombin was not identified. Thrombin is a serine protease known to be synthesized in the liver. It plays a central role in hemostasis by converting plasma fibrinogen into fibrin and by promoting platelet aggregation via proteinase-activated receptor (PAR) activation. In addition to its coagulation pathway role, thrombin influences a number of pathophysiological processes, including inflammation, tissue repair, angiogenesis, and tumor invasion[4,12–14]. While the reported increased presence of thrombin in inflamed mucosa from inflammatory bowel disease patients could easily be explained by the bleeding associated with tissue damage[11], it was more surprising to detect active thrombin in tissues from healthy individuals. One of the objectives of the present study was therefore to investigate the cellular source of thrombin in the gut mucosa.

Intestinal microbiota naturally grows as a mucus-coated polymicrobial community, embedded as a biofilm organization, separated from the epithelial surface by sterile mucus layer[15–19]. Nevertheless, biofilm encroachment at the surface of mucosal tissue is one of the most relevant drivers of persistent bacterial infections, thus constituting a major challenge for human and animal health[20]. Not only have such biofilms been associated with disease, they contribute directly to the foundation of inflammatory-based mucosal injuries[21–24]. Given the importance of spatial segregation between commensal bacteria and the epithelia, we postulated that the epithelium uses specific mechanisms to prevent the formation of a deleterious microbiota biofilm blanket in contact with host tissues.

We identify here the presence of thrombin in intestinal epithelium and its constitutive release on the apical side in physiological conditions. We show that epithelial thrombin is regulated by the presence of microbiota, and further that thrombin exerts a protective role at mucosal surface. Thrombin maintains mucosal homeostasis via its ability to cleave microbiota biofilm-derived proteins, thereby preventing biofilm contact with tissues, and limiting bacterial invasion.

## Results

**Intestinal epithelium releases constitutive active thrombin.** Reverse transcription analysis showed the presence of thrombin mRNA in intestinal epithelium from healthy human colon crypts (HC) as well as in three different human intestinal epithelial cell lines (Caco-2, HT-29, SW480, Fig. 1a, Supplementary Table 1).

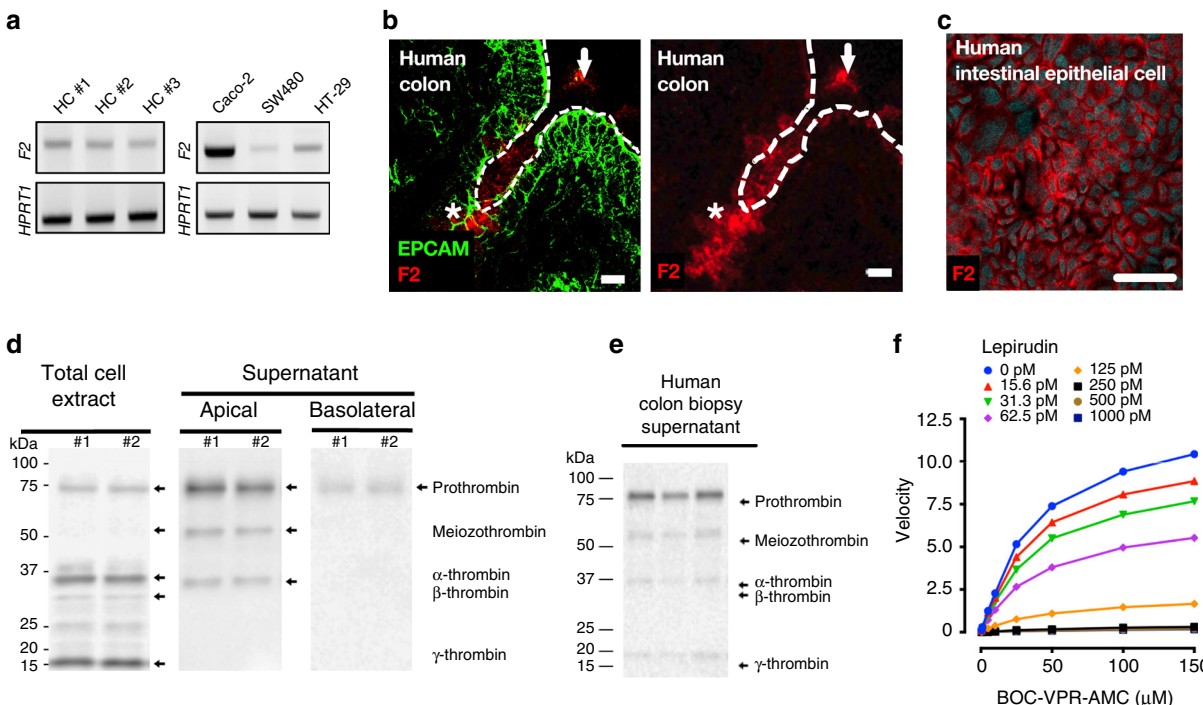

**Fig. 1** Intestinal epithelium releases active thrombin in gut lumen. **a** *F2* mRNA transcripts (Factor 2, thrombin) were detected in colonic crypt epithelium from three different healthy human (left), and three different human cancer cell lines (Caco-2, SW480 and HT-29, right). Representative blots from 3 independent experiments. **b** Immunostaining of Epithelial cell adhesion molecule (EPCAM, green, epithelial cell marker) and thrombin (red) in human colon biopsy displays thrombin-expressing epithelial cells (stars) and secreted thrombin in the lumen (arrows). Images are representative of three human donors. Scale bar corresponds to 20 μm. **c** Immunostaining of thrombin (purple) and nuclei (cyan blue, 4′,6-diamidino-2-phenylindole or DAPI) confirmed the specific staining of thrombin in Caco-2 cells. Images are representative of at least 3 independent fields. Scale bar corresponds to 50 μm. **d** Western blots revealed the presence of prothrombin (72 kDa) and its active isoforms (50, 32, 28 and 15 kDa) in Caco-2 cell protein extract (left blot). Released thrombin quantity was higher in apical (middle blot) compared to basolateral media (right blot). Representative blots from 3 independent experiments. **e** Western-blot confirmed the presence of human prothrombin and its active isoforms in colonic biopsy supernatant from three healthy donors. Representative blot from 3 independent experiments **f** Caco-2 cultured for 24 h in serum-free medium releases a trypsin-like activity (boc-VPR-AMC fluorescent substrate) that is dose-dependently inhibited by the thrombin inhibitor (lepirudin). Activity assay was reproduced in 3 independent experiments

Immunofluorescence detection of thrombin in healthy human colonic biopsies illustrated the expression of thrombin protein in the intestinal epithelium (stars), and in the lumen (arrows, Fig. 1b). Thrombin protein expression was confirmed in the human epithelial cell line (Caco-2) by immunohistochemistry (Fig. 1c). Western-blot analysis of cell extracts and supernatants from Caco-2 cells grown on transwells; confirmed the presence of released thrombin, both in its inactive proform (72 kDa band) and in its active forms (meizothrombin at 50 kDa, thrombin alpha at 32 kDa, thrombin beta 28 kDa, thrombin gamma 15 kDa, Fig. 1d). Most of the thrombin protein was released on the apical side of Caco-2 monolayers grown in transwells, only discrete bands were detected in Caco-2 basolateral supernatants (Fig. 1d). Western-blot analysis also confirmed the presence of thrombin in supernatants from incubated human colon biopsies harvested from healthy donors (Fig. 1e). Both the proform and active forms of thrombin were found in human colon biopsy supernatants, mucus scraped at the surface of colon mucosa, fecal samples from healthy human volunteer and naïve mice (Supplementary Figure 1). Proteolytic activity released by unstimulated Caco-2 cells (24 h) was concentration-dependently inhibited by lepirudin, a specific inhibitor of thrombin, further demonstrating that epithelial cells release active thrombolytic activity in the range of $50 \, \text{mU} \cdot \text{ml}^{-1}$ (Fig. 1f). Thrombin detection in human colonic tissues did not co-localize with Thioflavin T staining, indicating that in physiological conditions, mucosal thrombin is not associated with amyloid fibrin protein aggregates (Supplementary Figure 2). Overall, these data demonstrate that intestinal epithelial cells are a local source of active thrombin, released and active mostly on the luminal side of intestinal mucosa.

**Thrombin regulation in intestinal epithelium.** Prothrombin activation is known to be tightly regulated by the prothrombinase complex, composed of heterodimers of coagulation factors 10a and 5a. Because active thrombin was detected in supernatants from intestinal epithelial cell line (Caco-2, Fig. 1f), we investigated the possible presence of transcripts from the pro-thrombinase complex *F5* and *F10* genes in two human intestinal epithelial cell lines (Caco-2 and HT-29) as well as in isolated human colonic crypts. In all cases, transcripts from *F5* and *F10* genes were detected (Fig. 2a), showing that intestinal epithelial cells possess all the machinery required for the production of active thrombin. Further, we confirmed the presence of the F10 and F5 proteins in cell lysates of Caco-2 (Fig. 2b). Finally, in the presence of the prothrombinase F10-specific inhibitor Apixaban (0.01, 0.1 and 1 µM), thrombin activity produced by intestinal epithelial cells was significantly inhibited, proving that pro-thrombinase complex is present in intestinal epithelial cell cultures and is involved in active thrombin generation (Fig. 2c).

Because most of the thrombin protein produced by unstimulated intestinal epithelial cells was released at the apical side, we postulated that thrombin expression might be regulated by luminal factors, and potentially the presence of microbiota at the epithelial surface. We therefore investigated the mucosal expression of thrombin in the colons of germ-free mice. Average threshold cycle was 16 for the liver and 28 for colon mucosa. We

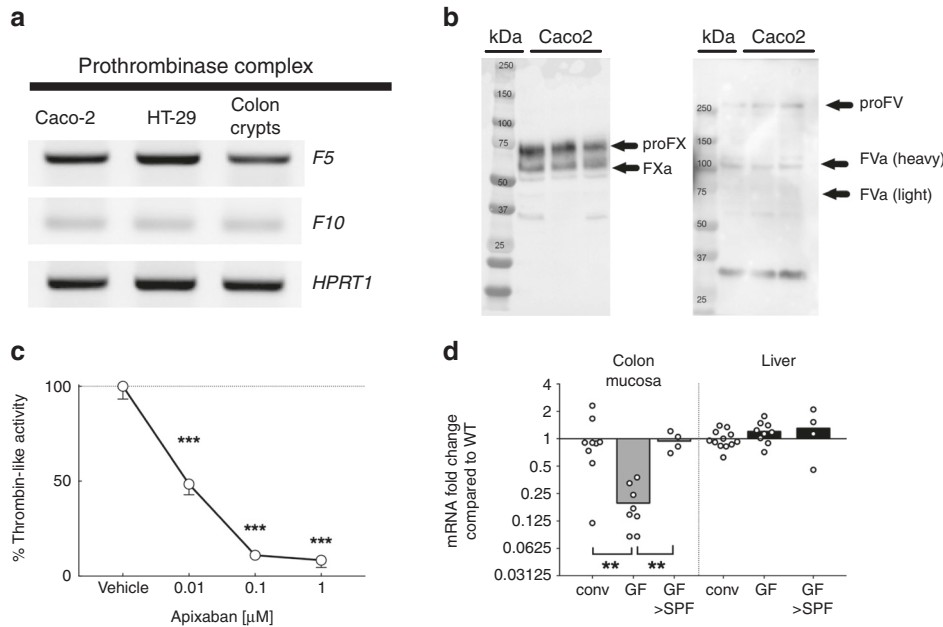

**Fig. 2** Thrombin is activated by epithelial-prothrombinase complex and is regulated by commensal microbiota. **a** mRNA transcripts for the two genes of the prothrombinase complex (gene *F5* and gene *F10*) were detected in two different human cancer cell lines (Caco-2 and HT-29) and isolated human colonic crypts. The same samples used in Fig. 1a to detect *F2* transcripts were used to detect F5 and F10 transcripts in Caco-2 and HT-29. Representative blots from 3 independent experiments. **b** Western blots revealed the presence of Factor 10 (inactive, 74 kDa and active form 54 kDa, left blot) as well as Factor 5 (inactive, 300 kDa; active heavy chain 94 kDa and active light chain 74 kDa, right blot) in Caco-2 cell protein extract (left blot). Representative blots from 2 independent experiments. **c** Epithelial thrombin activity produced by Caco-2 was dose-dependently inhibited by the presence of the F10 inhibitor Apixaban during 24 h (0.01, 0.1 and 1 µM). Activity assay was reproduced in 3 independent experiments. Data are presented as mean ± SEM. One-way ANOVA with Fisher's LSD test ***$P < 0.001$ vs. vehicle group (**d**). Germ-free mice (GF, $n = 5$), germ-free recolonized with specific-pathogen-free (SPF) cecal content (GF > SPF, $n = 5$) and conventionally-bred mice (CONV, $n = 8$) were sacrificed at 10–14 weeks old. Relative expression of *F2* mRNA transcript (thrombin) was unchanged in germ-free mouse liver compared to conventionally-bred mouse liver and GF > SPF. In germ-free mouse, *F2* mRNA was significantly reduced in colon mucosa compared to conventionally-bred mouse colon mucosa. GF recolonized with SPF microbiota had identical transcription of epithelial thrombin compared to conventional mice. One-way ANOVA with Fisher's LSD test **$P < 0.01$ vs. GF group. Data are represented as scatter plot with the mean bar

observed that thrombin mRNA was significantly reduced (by 70%) in germ-free mice, compared to the levels detected in conventionally-bred mice (Fig. 2d). Liver expression of thrombin was not different in germ-free compared to conventional mice (Fig. 2d). Interestingly, mucosal transcription of thrombin mRNA was completely restored in germ-free mice 3 weeks after recolonization with pathogen-free standard microbiota (cecal content) (Fig. 2d). These results confirmed the constitutive production of active thrombin in intestinal epithelium, and its control by the presence of microbiota.

**Constitutive thrombin activity preserves mucosal homeostasis**. To determine the physiological role of thrombin at mucosal surfaces, we inhibited its activity, by administering the irreversible thrombin-specific inhibitor, lepirudin, intracolonically to mice[25]. After ten daily administrations (10 µg per day per mouse), we observed a significant increase of macroscopic damage compared to animals receiving intracolonic administration of vehicle (NaCl 0.9%) (Fig. 3a). Histological analysis also revealed significant damage in the distal colon of animals treated with lepirudin (Fig. 3b). The histological damage consisted of crypt elongation, area of goblet cell depletion, as well as neutrophil transmigration into the lumen (Fig. 3c). We performed a transcriptomic qPCR analyses of genes involved in host-microbiota related functions (*Defb4, Tff3, Reg3g Reg4b, Camp, Muc2*), tight junctions (*Zo1, Ocln, Cldn1, Cldn2, Cldn5*) as well as inflammatory markers (Co*x2, Nos2, TNFa, IFNg, IL17A, Adgre1, Cxcl1, IL1b*) in the distal colon of vehicle- and lepirudin-treated animals

(Supplementary Figure 3). We used a principal coordinate analysis (PCoA) with Bray-Curtis dissimilarity matrix to show overall distance in each mouse transcriptome (Fig. 3d). As illustrated by PcoA, but also by hierarchical clustering dendrogram (Supplementary Figure 3A), there was a significant shift between the transcriptome of vehicle- and lepirudin-treated groups (Permanova $P$ value = 0.0367). The strongest discriminants in this shift were group of genes involved in host-microbiota interactions (*Camp, Muc2* and *Tff3*; Supplementary Figs. 3 and 4) as well as the inducible nitric oxide synthase (*Nos2*). This separation caused by lepirudin failed to reach significance for inflammatory genes (Permanova $P$ value = 0.0949, Supplementary Figure 3C) or tight junction genes (Permanova $P$ value = 0.553, Supplementary Figure 3D). Together, these results demonstrate that inhibition of epithelial thrombin activity in the lumen is sufficient to cause intestinal injuries and to cause alterations in the host-microbiota related transcriptome.

**Impact of constitutive thrombin activity on gut microbiota**. To determine whether constitutive thrombin activity would change the composition and relative abundance of gut microbiota, we sequenced the 16 S rDNA V3–V4 regions from fecal samples of mice at day 0 and after 10 days treatment with lepirudin (Supplementary Figure 5A). Principal component analysis revealed that the shift induced by the intracolonic administration of the thrombin inhibitor lepirudin was modest as group's convex hulls were largely superimposed (Supplementary Figure 5B). However, the cladogram obtained by LEfSe (Linear discriminant

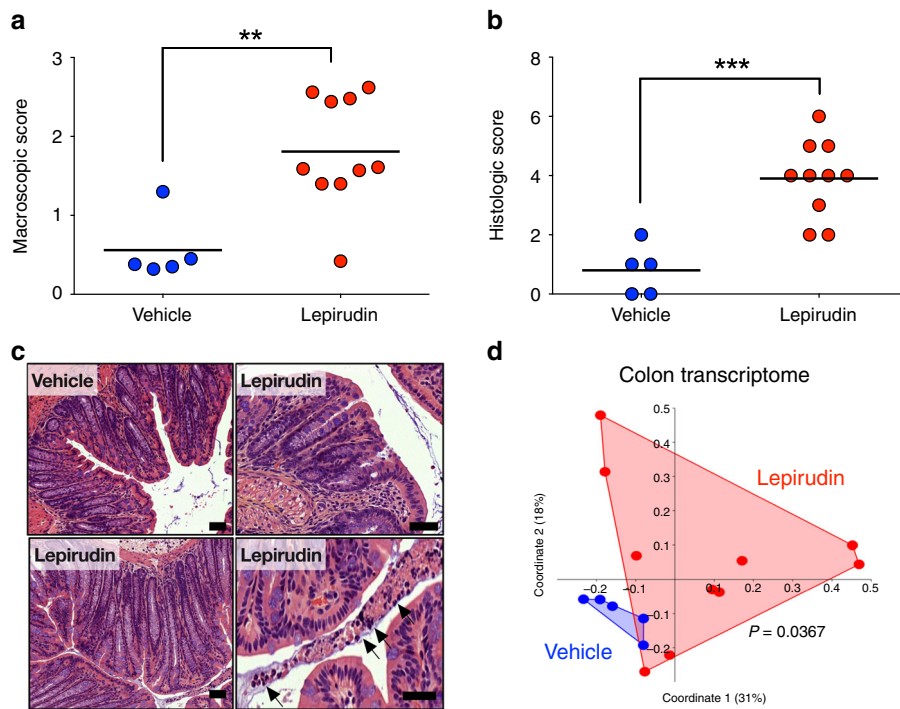

**Fig. 3** Intestinal homeostasis is compromised by inhibition of thrombin activity in colon lumen. C57Bl/6 mice were treated daily with either vehicle ($n = 5$) or lepirudin ($n = 10$, 10 µg/day) via intracolonic route, euthanized after 10 days and used for further analysis (**a**–**d**). **a** Macroscopic (Wallace scoring endpoint[58]) and **b**. Microscopic damage scores[49] in lepirudin-treated group were significantly greater than the scores measured in vehicle controls. **a**, **b** Data are represented as scatter plot with the mean bar. Unpaired Mann–Whitney two-tailed test ***$P < 0.001$ **c** Hematoxylin-eosin images revealed a normal histology in vehicle-treated group, while in lepirudin-treated group, we observed crypt elongation (lower left panel), areas with goblet cell depletion (upper right panel), as well as neutrophils transmigration in the lumen (lower right panel, arrows). Scale bars represent 50 µm. Representative images of vehicle ($n = 5$) and lepirudin-treated ($n = 10$) group are reported. **d** Transcriptome analysis of *Muc2, Camp, Tff3, Defb4, Reg3g, Reg3b, Cox2, Nos2, Tnf, Ifng, Cxcl1, IL17a, Adgre1, IL1b, Cldn1, Cldn2, Cldn5, Tjp1 Zo1* and *Ocln* genes was performed by qPCR. Principal coordinate analysis with Bray-Curtis dissimilarity matrix illustrates distances between each mouse transcriptome (each dot represents an individual mouse transcriptome), and demonstrate a significant separation between vehicle and lepirudin-treated group. PERMANOVA Bonferroni corrected $P = 0.0367$

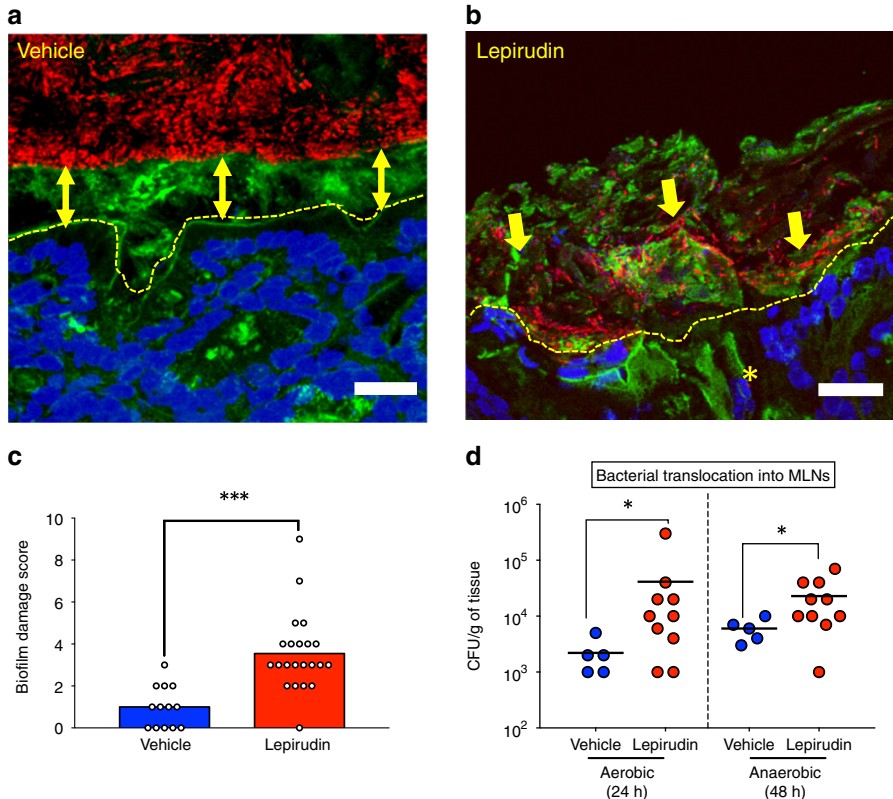

**Fig. 4** Suppression of basal thrombin activity modifies spatial organization of microbiota biofilm. C57Bl/6 mice were treated daily with either vehicle ($n =$ 5) or lepirudin ($n = 10$, 10 µg per day) via intracolonic route, euthanized after 10 days. **a, b** Distal colon sections were Carnoy's fixed. All bacterial cells were labeled with the universal probe Eub338 for fluorescent in situ hybridization (red color). Wheat germ agglutinin was used to stain the polysaccharides-rich mucus layer on these samples (green color). All images were counterstained with the nuclear stain, 4',6-diamidino-2-phenylindole DAPI (blue color). **a** In vehicle-treated colon, double-arrows highlights the presence of sterile mucus layer separating the colon epithelium (dashed line) from the dense microbiota biofilm. **b** In contrast, lepirudin-treated animals were characterized by a complete disorganization of microbial biofilms and mucus layer in distal colon. Adherent and isolated microcolonies were evident (arrows) and rods were visible in submucosal tissue (asterisks). Images are representative of $n =$ 5 mice. Scale bars represent 20 µm. **c** Abnormal alterations to gut microbiota biofilm organization were blindly recorded and demonstrated an increased damage score in animals treated with lepirudin compared to vehicle control groups. Data are represented as scatter plot with the mean bar. Unpaired Mann–Whitney two-tailed $t$ test ***$P < 0.001$. **d** Bacterial translocation, of aerobes and anaerobes, into mesenteric lymph nodes (MLN) was significantly greater in animals treated with lepirudin compared to vehicle-treated animals. Data are represented as scatter plot with the mean bar. Unpaired Mann–Whitney two-tailed $t$ test *$P < 0.05$

analysis effect size) analysis showed a lepirudin-specific microbial signature characterized by a higher abundance of the genus *Barnesiella* (belonging to the phylum Bacteroidetes and family *Porphyromonadaceae*), compared to the vehicle-treated group (Supplementary Figure 5C).

As the physiology of complex microbial communities is strongly dependent on the immediate surroundings of each microbe, we then asked if thrombin would alter the microbial microenvironment. Fluorescent in situ hybridization was used to visualize spatial organization of microbiota biofilm. In animals treated with vehicle, no mucosa-adherent bacteria were observed, and no bacteria were detected deep in the crypts, within the epithelial cells or in the submucosa. Microbiota in the distal colonic mucosa was clearly separated from the epithelial surface by a dense mucus layer, free of bacteria, although high density microbial biofilms were obvious in areas where mucus contacted fecal material (Fig. 4a). Intestinal biofilms in animals treated with lepirudin had a strikingly different spatial organization from that of vehicle-treated mice (Fig. 4b), resulting in an overall higher biofilm damage score (Fig. 4c, Table 1). Short and long rod bacteria could be seen forming microcolonies or partially segregated from each other at the outer edge of the mucus layer

(when present) and feces. These microcolonies were largely encased in a loose polysaccharide-rich matrix (Fig. 4b). Lepirudin allowed isolated planktonic bacteria to colonize the submucosal tissues (yellow star, Fig. 4b). As this result indicated a possible breach of the mucosal barrier, we investigated whether commensals were able to translocate across the mucosa to distant organs such as mesenteric lymph nodes. The results show that the numbers of both aerobic and anaerobic bacteria were significantly increased in the mesenteric lymph nodes of mice treated with intracolonic administration of the thrombin inhibitor, lepirudin, compared to animals treated with the vehicle alone (Fig. 4d). Overall, the findings indicate that while ablation of mucosal thrombin activity modestly alters microbiota community composition, it does create opportunities for microbial invasive behavior. These results suggest that constitutive thrombin activity at the surface of intestinal mucosa exerts a constraining role on mucosal biofilms, that prevent its contact with the epithelial surface.

**Thrombin alters human gut microbiota biofilm structure.** To further explore if and how thrombin could contribute further to

**Table 1 Biofilm damage score of the distal colon microbiota**

| | Biofilm damage score | | | |
| --- | --- | --- | --- | --- |
| | **0** | **1** | **2** | **3** |
| Inner mucus layer invasion | Rare | Few | Numerous | Numerous and dense colonies |
| Biofilm distance to epithelia | >20 μm | >10 μm | Numerous contact | Dense biofilm in contact |
| Biofilm density | High | Mild | Scattered | Mostly planktonic |
| Bacterial translocation into lamina propria | None | Few | Numerous | Dense colonies |
| Immune cell transmigration into the biofilm | None | Scattered | Dense | Dense and extracellular DNA |
| Bacteria morphology | More than 5 morphotypes | <4 morphotypes | <3 morphotypes | Predominance of one morphotype |

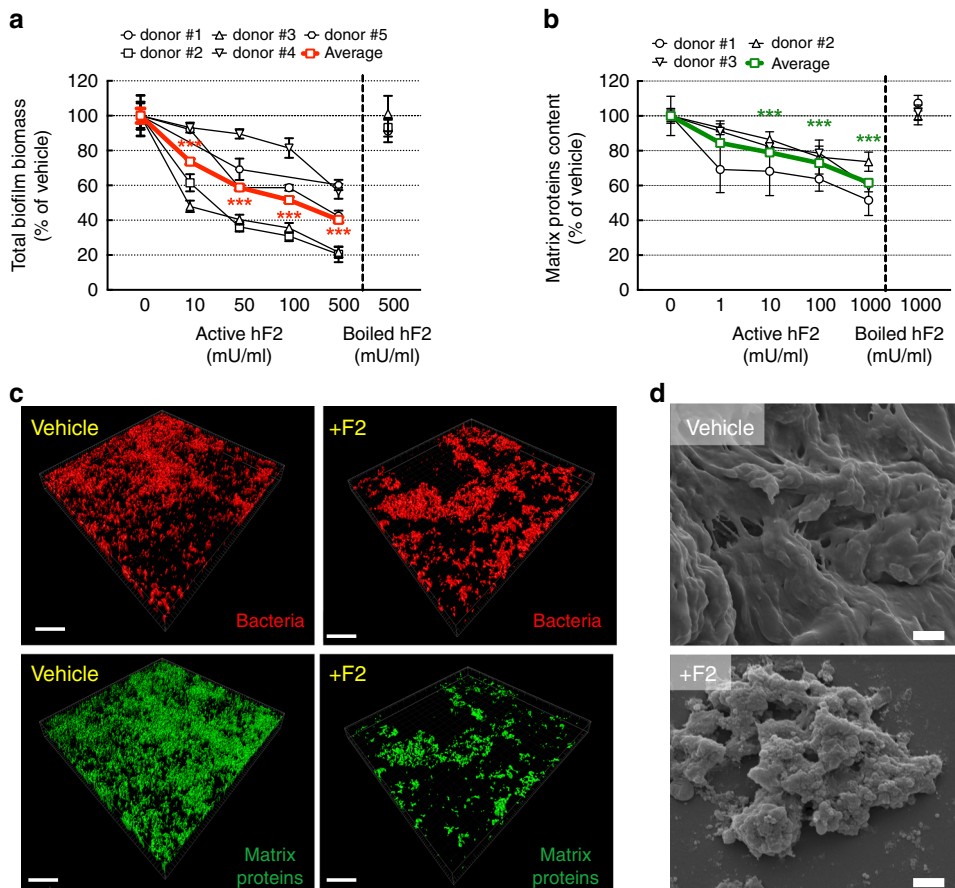

**Fig. 5** Thrombin alters multispecies human biopsy microbiota anaerobic biofilms. Multispecies anaerobic biofilms from 5 different human healthy donors were generated *ex-vivo*. Mature biofilms were then exposed to various concentrations of human active thrombin or inactive boiled thrombin for 24 h. **a** Active thrombin, but not boiled thrombin, dose-dependently reduced total biofilm biomass (crystal violet assay), (**b**) as well as reduced the total content of matrix-associated proteins (FilmTracer Ruby matrix biofilm stain). A & B Data are presented as mean ± SEM. One-way ANOVA with Fisher's LSD test vs. control group without thrombin *$P < 0.05$, **$P < 0.01$, ***$P < 0.001$. Each concentration with >12 biofilms per donor, $N = 3$ independent experiments. **c** Representative confocal 3D-surfaces reconstructions from human biofilms confirmed a strong effect of thrombin (100 mU ml$^{-1}$, equivalent to 1 nM, 24 h) on matrix-associated proteins (green color), while total adherent bacteria staining (propidium iodide, in red) was marginally affected. Scale bars represent 20 μm. Representative images from at least 3 independent experiments. **d** Scanning electron microscopy was performed on the same human biofilm treated or not with thrombin (100 mU ml$^{-1}$, equivalent to 1 nM, 24 h). Biofilms from vehicle-treated group were fully covered with a thick matrix slime with unnoticeable bacteria (upper panel). Biofilm treated with thrombin had an altered matrix structure, bacteria cells becomes clearly visible (lower panel). Representative images from $n = 3$ healthy donor's biofilm. Scale bars represent 1 μm

the spatial segregation of commensal biofilms, we first cultured mucosa-associated microbiota, from 4 healthy human colon biopsies, in the presence of active human thrombin in liquid media. After 15 h exposure under these specific conditions of incubations, no direct bactericidal or bacteriostatic property was detected upon thrombin exposure (Supplementary Figure 6). We then reconstituted mucosa-associated human microbiota under its natural biofilm phenotype in anaerobic conditions. Mature human biofilms were generated for 48 h and were then exposed for an additional 24 h to active or inactivated human thrombin. Biofilm biomass (cells and total matrix-associated content) was significantly reduced concentration-dependently, starting from 10 mU ml$^{-1}$ of active thrombin (Fig. 5a). Boiled thrombin (500 mU ml$^{-1}$) did not cause such an effect (Fig. 5a), nor did

thrombin in the presence of its specific inhibitor lepirudin (Supplementary Figure 7). At concentrations lower than 100 mU ml$^{-1}$, thrombin had no effect on bacterial viability within biofilms (Supplementary Figure 8A). Starting at concentrations of 50 mU ml$^{-1}$, active human thrombin increased the dispersion of live biofilm-derived bacteria (Supplementary Figure 8B). To understand further the origin of biofilm biomass reduction, we stained matrix-associated proteins (SYPRO biofilm matrix) and polysaccharides (wheat germ agglutinin). We then extracted the biofilm matrices by disruption of electrostatic interactions using 1.5 M NaCl buffer (pH 7.4), and measured the amount of fluorescently labeled matrix-associated proteins and polysaccharides. We found that thrombin, starting at 1 mU ml$^{-1}$, significantly reduced total content of matrix-associated proteins (Fig. 5c). At higher concentrations (>100 mU ml$^{-1}$), thrombin reduced the total polysaccharide content in biofilms (Supplementary Figure 8C and 8D). Scanning electron microscopy revealed bacterial cells hidden beneath the dense and fully covering matrix slime in untreated human biofilms (Fig. 5d, upper panel). When a biofilm from the same patient was treated with human thrombin (100 mU ml$^{-1}$ for 24 h), individual bacteria became visible as the matrix was severely damaged (Fig. 5d, lower panel). Further, we have investigated the molecular mechanisms by which active thrombin (100 mU ml$^{-1}$ for 24 h) cleaves matrix-associated proteins by using N-terminomics/TAILS (Terminal Amine Isotopic Labeling of Substrates, identification of endoproteolysis site)[26,27] and shotgun proteomics analysis (Supplementary Figure 9). The list of biofilm-peptides processed by human thrombin including the cleavage site positions are shown in Supplementary Table 2, together with the putative protein corresponding to the cleaved peptides and the microbial species known to express such peptides. Overall, these data provide evidence that human thrombin can damage mature human multispecies biofilms, through enzymatic processing of selective protein constituent of the biofilm matrix backbone.

**Thrombin is expressed in skin, lung, bladder and ileon**. As thrombin (*F2* gene) appeared to be important for intestinal mucosa homeostasis by constraining microbial biofilms, we hypothesized that epithelia from all major host-microbiota surfaces can also produce thrombin for the same purpose. We detected *F2* mRNA in epithelial cell lines derived from intestine, skin and lung, in crypt epithelium and bladder urothelium harvested from healthy human tissues (Fig. 6a) as well as from C57Bl/6 mice (Fig. 6b). We sequenced the specific *F2* amplicon (297 bp for human, 180 bp for mouse), and performed a blast alignment revealing a > 99% homology with the respective human and mouse *F2* genes, confirming an active transcription of the thrombin gene. These data further expand our knowledge on the presence of active thrombin in epithelial organs, which at least in the intestine is under the direct regulation of microbiota. Our data thus point to a previously unknown role for thrombin epithelia-biofilms interactions.

## Discussion

The present findings demonstrate that thrombin, a serine protease classically involved in the coagulation cascade and known to be produced in the liver, can originate from intestinal epithelium, where it can in principle, contribute to mucosal protection by

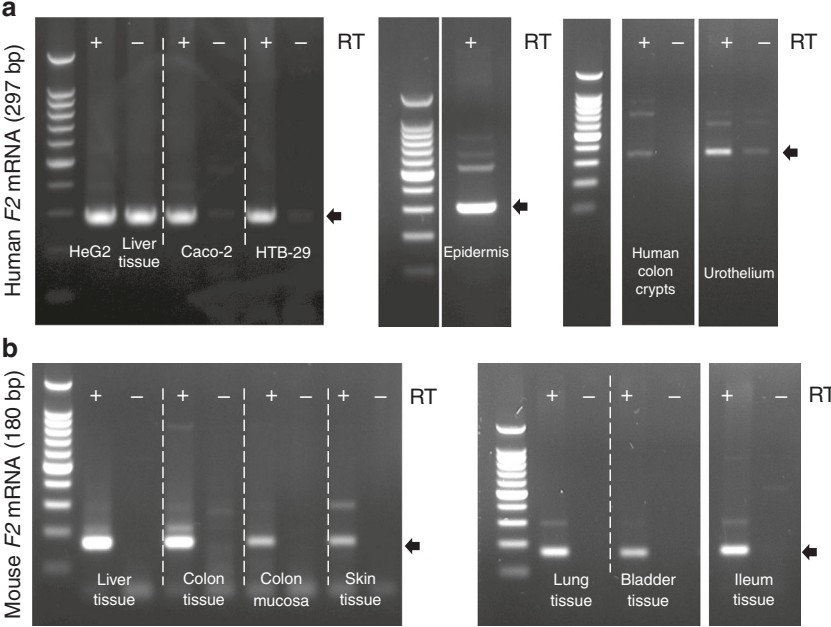

**Fig. 6** The *F2* gene encoding thrombin is expressed in all major epithelia under healthy conditions. *F2* mRNA transcript (thrombin) was detected by reverse transcription PCR in (**a**) human and (**b**) C57Bl/6 mouse epithelia. **a** *F2* mRNA is present in lane RT + corresponding to human hepatocyte cell line (Hep G2), liver tissue, intestinal epithelial cell line (Caco-2), lung epithelial cell line (HTB-29), healthy skin epidermis, human isolated epithelial cells from colon crypts and primary culture of urothelium. Lanes RT- are negative control for amplification of genomic contamination. Small gaps represent the same gel from which irrelevant lanes were cut out. Larger gaps correspond to three different gels, PCR conditions were 40 cycles at 60 °C for HepG2, Caco-2, HTB-39 and 35 cycles at 62 °C for epidermis, colon crypts and urothelium. Amplicon at 297 bp (arrow) was sequenced and confirmed to be human *F2* mRNA. **b** *F2* mRNA is present in lane + RT corresponding to liver tissue, colon tissue, colon mucosa, skin tissue, lung tissue, bladder tissue, and ileum tissue from mouse. Lanes -RT are negative control for amplification of genomic contamination. Small gaps represent the same gel from which irrelevant lanes were cut out. Larger gaps correspond to two different gels, PCR conditions were 40 cycles at 60 °C for liver, colon tissue, colon mucosa, skin and 35 cycles at 62 °C for lung, bladder and ileum tissue. Amplicon at 180 bp (arrow) was sequenced and confirmed to be mouse *F2* mRNA. RT-PCR experiments have been reproduced at least twice independently

maintaining healthy spatial segregation between the host and the microbiota.

Adding to our current report, ectopic expression of thrombin (both mRNA and protein) has been suspected in the epithelium from benign and malignant prostate tumors[28], in brain endothelial cells[29], as well as in epithelial-like cells that form ovarian follicle granulosa cells[30]. In our study, we detected unequivocal expression of thrombin in non-tumoral intestinal, lung, skin and bladder epithelium. Furthermore, we demonstrated that the epithelial production of thrombin mRNA is directly controlled by microbiota colonization in the intestine. Interestingly, thrombin mRNA production in liver remained unchanged in germ-free mice, suggesting that the intestinal epithelium possesses specific regulatory mechanisms for the local production of thrombin in the digestive tract.

Although anticoagulants and direct thrombin inhibitors administered orally are increasingly used for treatment of chronic cardiovascular pathologies[31], their safety remains a concern. Patients commonly suffer from gastrointestinal adverse effects that may be severe and even fatal in the case of gastrointestinal bleeding[25,32]. Our results demonstrated a protective role of constitutively produced epithelial thrombin. It thus can be hypothesized that chronic inhibition of epithelial thrombin activity by oral anticoagulant treatments could possibly lead to uncontrolled and invasive biofilm phenotypes, initiating tissue damage. Indeed, in Supplementary Figure 10, mice were orally treated with dabigatran etexilate, a direct thrombin inhibitor used in humans for cardiovascular pathologies, or with warfarin, a vitamin K antagonist. We observed in these mice gastrointestinal bleeding (presence of blood in feces), tissue damage in stomach (petechia, purpura, edema), and bacterial translocation to mesenteric lymph nodes.

In addition to its transcriptional regulation, inactive prothrombin has to be cleaved into active thrombin. This activation can be achieved not only by autoproteolysis, but also by the activity of many other proteases of the coagulation cascade. Among the many different possible ways to activate thrombin, we investigated the concurrent presence along with thrombin, of the prothrombinase complex composed of Factors 5a and 10a. Both factors are present (both mRNA and protein) and biologically active in the intestinal epithelium to transform inactive to active thrombin. Further, we found that thrombin activity is under the control of the prothrombinase complex, as Apixaban, an inhibitor of this complex, was able to inhibit the release of thrombin activity from intestinal epithelial cells. Pro-thrombin activation also requires the vitamin K-dependent carboxylase (specifically the VKORC1 subunit)[33] known to be produced by commensal microbiota. Germ-free mice reach an euthanasia endpoint when fed an irradiated AIN-76A diet, if not supplemented with vitamin K[34]. Of note, vitamin K deficiency in humans resulting from broad-spectrum parenteral antibiotic treatment, is associated with severe gastrointestinal damage, such as bleeding and perforated gastric ulcers[35]. Here we propose that thrombin inhibition in the lumen of distal colon might directly predispose to mucosal injuries, at least in part due to microbiota biofilm adhering to mucosal surfaces. In addition to this previously unknown role for thrombin in microbiota organization, other pathophysiological roles for epithelial thrombin are plausible, but yet unexplored. More research is needed to determine whether constitutive expression of epithelial thrombin might also regulate epithelial biology, potentially by protease-activated receptors activation.

On the basis of our results, a role for thrombin activity in the modulation of *Barnesiella* abundance is suspected, although thrombin's effect on shaping overall microbiota composition is likely to be minor. Under inflammatory conditions, human neutrophil elastase is able to cleave out small C-terminal highly cationic fragments from thrombin, which in turn exert bactericidal effects on isolated pathogens[36,37]. Adding to these reports, our data suggest that full-length active thrombin, exerts a specific biofilm-disrupting activity that depends on its proteolytic function. Together, thrombin seems to have a dual action on microbes: to segregate host mucosa from bacterial biofilms for the full-length active thrombin, and bactericidal effects on planktonic bacteria for truncated, proteolytically inactive thrombin. Interestingly, western-blot analysis demonstrated that different forms of thrombin (pro-form, active form, truncated forms) are present at the epithelial surface. Each form might exert diverse effects on bacteria in vivo, adding to the impact of proteolytically active thrombin on bacterial biofilm biomass.

Biofilm bacteria are embedded in a protective matrix having a complex composition (e.g., RNA/DNA, proteins, polysaccharides). The development of this biofilm organization constitutes a major process directly relevant to human and animal health[20]. Therefore, the prevention of biofilm overgrowth and disruption of already established deleterious biofilms is crucially important. Destroying the biofilm matrix backbone, for example via enzymatic lysis, seems to be an interesting approach for biofilm eradication. Several microbially-derived enzymes have indeed been reported to degrade components of bacterial biofilm matrix, although these reports relied exclusively on monospecies biofilms[38–42]. Our study adds a previously unsuspected component of biofilm matrix regulation: active thrombin produced by the host epithelial mucosa. Epithelial-derived thrombin can thus be added to the list of matrix-degrading antibiofilm agents, with a strong effect on biofilms originating from complex multispecies human microbiota. Our study suggests that epithelial-derived proteolytic factors are produced as a physiological response towards microbiota biofilms growing at the mucosal surface of the digestive tract. Our demonstration that thrombin is also expressed by epithelia from lung, bladder and skin, where the same role for thrombin could be expected, further suggests that this novel mechanism may be a target for new therapies to modify pathogenic biofilm encroachment at these host-microbial surfaces.

We determined that the concentration of thrombin released by monolayers of intestinal epithelial cells was in the range of $50\ mU\ ml^{-1}$. We used a similar thrombin concentration for in vitro experiments on biofilms. Of note, this thrombin concentration range is very low compared to the systemic blood thrombin activity required to induce coagulation (in the range of $10\ U\ ml^{-1}$)[43]. Although we report a protective function for basal thrombin activity, where low concentrations of epithelial thrombin could be beneficial to control mucosal biofilms and avoid their direct contacts with tissue, increased thrombin activity is also associated with chronic inflammation of the gut in cases of inflammatory bowel disease patients[11]. In that case, one can hypothesize that these relatively high concentrations of thrombin at mucosal surfaces might be detrimental, fragmenting and dispersing biofilms to cause tissue damage. Indeed, an inflamed area of the colon is associated with an overall destruction of normal biofilm organization, with heterogeneous morphology, ranging from isolated cells to clusters of epithelia-adherent biofilms[15,18,22,23,44]. Further studies related to the role of epithelial-derived thrombin in pathologies, and to the control of thrombin expression and activity, are thus warranted.

The biofilm phenotype of intestinal microbiota has been well established in the healthy digestive tract and is conserved throughout the animal kingdom[15,17,19,21,45]. The ability to maintain such biofilms that distance the microbiota from host tissue is likely of evolutionary significance for digestive health as well as for host survival. We propose that epithelial thrombin helps maintain a spatial segregation between the microbiota

biofilm and the host. The present findings can have a direct impact on human health in view of the link between biofilms abnormal overgrowth on mucosal surface, in the gut, and beyond (e.g., inflammatory bowel diseases, colorectal cancer, cystic fibrosis, urinary tract infections and chronic wound ulcers)[20,46].

## Methods

**Human tissue collection.** Colon cancer resections from human donors were provided by the Centre Hospitalier de Toulouse (France). The Ethics Committee (Comités de Protection des personnes de la Région Occitane) approved the human research protocol (ClinicalTrials.gov Identifier: NCT01990716) and is compliant with all relevant European ethical regulations. Written and verbal informed consent was obtained before enrollment in the study. Biopsies used in the study were collected from macroscopically healthy area at distance from cancer tissue and when necessary transferred in sterile tubes for anaerobic transport (BBL, BD Bioscience). Tissue was manipulated under aseptic conditions and maintained throughout on ice and on PBS sterile buffer. Human crypts were isolated, cultured, and colon organoids were generated as described in[47], the absence of inflammatory cells in organoid cultures was confirmed (lack of CD45-positive staining on flow cytometry).

**Animals.** All animal procedures were approved by the Animal Care and Ethics Committee of US006/CREFE (CEEA-122, APAFIS#7762-20161125092278235) and are compliant with all relevant ethical regulations. C57BL/6 mice were kept in ventilated cages and acclimatized to the study conditions for 2 weeks before entering experiments. Mice used for experiments were 8 weeks old. Germ-free C57BL/6 mice (10–12 weeks old) generated by 2-stage embryo transfer and housed in flexifilm gnotobiotic isolators were obtained from McMaster University's Axenic Gnotobiotic Facility. All protocols were approved by the McMaster University Animal Care Committee and McMaster Animal Research Ethics Board (AREB) in an amendment to the Animal Utilization Protocol (#170836). Tissue samples from these mice were shipped to INSERM-IRSD.

In conventionally bred mice, thrombin inhibition was induced by daily intracolonic instillation of Lepirudin (10 μg, Bachem, Ki = 0.2 pM) under 3% isoflurane anesthesia. Control animals were treated with an equal volume of vehicle (0.9 % NaCl). After instillation, the animals were kept upside-down for 2 min. Animals were euthanized by cervical dislocation after 10 days of experimentation. The cumulative macroscopic damage was obtained by measuring colon thickness edema (in mm) and blind Wallace scoring endpoints[48]. Histological damage scoring was performed on formalin-fixed, paraffin-embedded sections stained with hematoxylin-eosin, and according to a previously published scoring system[49], on 5 fields per tissue section. Macroscopic and histologic scoring was blindly performed by skilled experimenters (one experimenter for macroscopic and two experimenters for histological scoring). Mesenteric lymph nodes were collected aseptically, weighed, homogenized and plated on Columbia blood agar (BD Biosciences) for 24 h for aerobes and 48 h for anaerobes (anaerobic jar) at 37 °C.

To investigate gastrointestinal damage associated with oral anticoagulants, one group of mice ($n = 12$) was treated orally with the direct thrombin inhibitor Dabigatran etexilate (50 mg/kg/day in mice corresponding to a human dose of ~4 mg/kg/day, SelleckChem, Euromedex, France). Another group ($n = 7$) was given warfarin in drinking water ad libitum (10 mg L$^{-1}$, Bristol-Myers Squibb, France). The dose based on water consumption intake was 2 mg/kg/day, which is equivalent to a human dose of 10 mg daily. Fecal score was recorded daily, based on i) fecal consistency (score 0 for normal feces, 1 for soft feces, 2 for diarrhea) and ii) the presence of blood in the feces (Hemoccult tests, score 0 for negative, score 1 for positive, score 2 for gross bleeding). We measured potential macroscopic lesions in the stomach according to the following criteria: no macroscopic damage (0), presence of purpura or petechia (1), tissue edema (1). Bacterial translocation to the mesenteric lymph nodes (presence of aerobic and anaerobic bacteria) was also assessed as described above.

**Cell culture.** Human intestinal epithelial cell lines (Caco-2 (ATCC HTB-37), HT-29 (ATCC HT-38), SW480 (ATCC-CCL-228)), human hepatocyte (Hep G2 from ATCC HB-8065), human lung epithelial cells (A549, ATCC CCL-185) were grown in DMEM high glucose GlutaMAX Supplemented with 1x non-essential aminoacids, 1x penicillin/streptomycin and 10% FBS (Gibco). Briefly, $3 \times 10^5$ IECs were plated on flat-bottom 6-well plates and grown for 7 days at 37 °C 5% CO$_2$ with culture medium replacement three times a week. Then, $1 \times 10^5$ Caco-2 cells were plated on polycarbonate 12-transwells and grown for 21 days, as described above. For cellular supernatants harvesting, cells were washed twice with PBS Ca$^{2+}$/Mg$^{2+}$-free and kept for 24 h in the cell culture medium described above, but without FBS.

**Gut microbiota biofilms.** For mucosa-associated biofilm reconstitution, human colonic biopsies were transferred in sterile tubes for anaerobic transport (BBL, BD Bioscience) just after collection in the endoscopy room. Colon biopsies were homogenized in a microtube pestle and mucosa-associated microbiota was first cultured overnight in rich anaerobe media (Wilkins-Chalgren broth, ThermoFisher

Scientific) supplemented with L-cysteine (5%, Sigma-Aldrich). Biofilms were generated using the Calgary Biofilm Device (Innovotech, Edmonton, Canada)[15,21,50,51]. All steps described herein were performed in anaerobic conditions in jar (Anaeropack, ThermoFisher Scientific). Mature biofilms (48 h) were transferred onto a new plate of minimal M63 media supplemented with glucose (2%), L-cysteine (0.5%) and containing various concentration of human active thrombin (thrombin from human plasma, maximal concentration of 1000mU ml$^{-1}$ was equivalent to 10 nM, Sigma T6884), thrombin inhibitor lepirudin (Bachem) or vehicle (phosphate buffered saline, pH 7.4 with 0.1 % bovine serum albumin) for 24 h. The biofilm biomass density and bacterial viability were determined respectively by crystal violet (1% in water, Reactifs RAL 317980) and rezasurin (2% in water, Sigma-Aldrich R7017)[15,21]. Matrix-associated proteins and matrix-associated polysaccharides (N-acetlyglycosamines and sialic acid) were stained respectively with FilmTracer Ruby matrix biofilm stain (ThermoFisher Scientific F10318) and wheat germ agglutinin (1/1000 dilution, Sigma L4895) for 2 h. Matrix was then extracted after incubation for 30 min in 1.5 M NaCl buffer, based on previously described method[52], and specific fluorescence was measured on fluorescent spectrophotometer (Tecan). Biofilm rate of dispersal was assessed by measuring the optical density (600 nm) and assessing colony-forming unit (CFU) of biofilm-dispersed planktonic bacteria recovered in the challenge plate. All results were expressed as percentages change of that of the means in the vehicle-treated group set as 100 %.

**Imaging of biofilms.** Biofilms were stained for 1 h, without fixation, with FilmTracer Ruby matrix biofilm stain (specific matrix-associated protein stain), fluorescein-labeled wheat germ agglutinin (matrix-associated polysaccharides stain) and propidium iodide (DNA-RNA stain, 1/1000 dilution, Invitrogen P1304MP) and visualized on a confocal microscope (Zeiss LSM 710). Biofilm damage score was blindly evaluated in 13 fields for vehicle group ($N = 5$ animals) and 22 fields in lepirudin ($N = 5$ animals), according to the criteria described in Table 1. Three-dimension surface rendering of stained biofilm was performed on Imaris Bitplane (v8, Concord, MA, USA). Alternatively, pegs containing treated biofilms were broken with needle nose pliers, and fixed in 2 % glutaraldehyde (Sigma-Aldrich) in 0.1 M Sorensen phosphate buffer (pH 7.4). Biofilms were dehydrated, dried by critical point drying (Leica EM CPD 300), and coated with 6 nm Platinium on a Leica EM Med 020 before being examined on a FEI Quanta 250 FEG scanning electron microscope, at accelerating voltages of 5 and 10 kV. FIJI freeware was used for final image mounting (v.1.51).

**Bacterial growth rate.** Human mucosa-associated microbiota were grown for 24 h in rich anaerobe media (Wilkins-Chalgren). Saturated cultures were then diluted in 96-well microplate to optical density of 0.1 (OD$_{600\ nm}$) in minimal M63 media supplemented with glucose (2%), L-cysteine (0.5%) and containing various concentration of human active thrombin (0 to 1250 mU ml$^{-1}$). The growth curves of each inoculum were generated from continuous OD$_{600nm}$ reading every 20 min for 15 h. Each value was expressed using means of duplicate experiment for each microbiota.

**Tissue imaging.** For fluorescent in situ hybridization (FISH), Carnoy's-fixed mice colon tissues were paraffin-embedded. Slides were hybridized with 1 μM of a universal bacterial 16 S fluorescent rRNA probe (EUB338-Cy3, 5′-GCTGCCTCCCGTAGGAGT-3′ Cy5, Eurofins) and counterstained for DNA by 4′,6-diamidino-2-phenylindole (DAPI, 1/1000 dilution Sigma-Aldrich) and polysaccharides content (wheat germ agglutinin labeled with fluorescein, 1/1000, Sigma-Aldrich L4895). Human colonic biopsies were cryopreserved in Optimal Cutting Temperature (OCT, Dako) and sectioned at 6 μm of thickness. Slides were thawed at room temperature for 20 min and blocked for 1 h (1% BSA, 0.3% Triton X-100, PBS 1×). Tissues were immuno-stained with goat polyclonal anti-thrombin (1/250 dilution, Santa Cruz, sc-16972) overnight, and double labeled with secondary antibody (anti-goat alexa-555, 1/1000 dilution Life Technologies A211432). Slides were counterstained with 4′,6-diamidino-2-phenylindole (DAPI, Invitrogen, France), Thioflavin T (0.001% dilution, Invitrogen, T3516) and/or fluorescein-labeled wheat germ agglutinin (1/1000 dilution, Sigma-Aldrich L4895). Epithelial cells were highlighted with anti-Epcam staining (1/200 dilution; Abcam, ab-32392). To reveal unspecific staining of thrombin antibody, isotype control (Santa Cruz, sc 2028) was used and incubated at the same concentration and under the same experimental conditions (Supplementary Figure 11). Representative images were obtained from blind acquisition of 4 different fields per animals. We acquired all images on Leica LSM 710 confocal microscope, and FIJI freeware was used for final image mounting (v.1.51).

**16S rDNA Sequencing.** Total DNA was extracted from feces at both baseline and after the treatment with lepirudin according to manufacturer's protocols (QIAamp DNA stool mini kit, Qiagen 51604) with slight modifications as described in ref.[53]. The 16 S bacterial rDNA V3–V4 regions were targeted by the 357wf-785R primers and analyzed by MiSeq at RTLGenomics (Texas, USA). An average of 11,000 (between 6896 and 15901) sequences was generated per sample. A complete description of the applied bioinformatic filters is available at www.rtlgenomics.com. Cladograms were drawn by the Huttenhower Galaxy web application

(huttenhower.sph.harvard.edu/galaxy/) via the LEfSe (Linear Discriminant Analysis Effect Size) algorithm[54].

**Transcription assays**. mRNA from intestinal epithelial cells, crypts and organoids were extracted by using the Nucleospin RNA/Protein kit (Macherey-Nagel). mRNAs from other human and animal tissues were extracted using the Qiagen RNeasy kit according to the manufacturer's instructions (Qiagen) and reversely transcribed into cDNA (iScript cDNA synthesis kit, Biorad). The PCR was performed on 384-well plates and on LightCycler 480 (Roche). The sequences used in the study are detailed in Supplementary Table 1. The expression levels of genes were normalized to both Glyceraldehyde 3-phosphate dehydrogenase (*GAPDH*) and hypoxanthine-guanine phosphoribosyltransferase (*HPRT*) as reference genes. Fold changes in the mRNA levels were calculated with the comparative $2^{-\Delta\Delta Ct}$ method. For RT-PCR blot analysis, 1 µg mRNAs were reverse transcribed using Maxima First Strand cDNA kit (Thermo Fisher). Subsequent PCR was performed on 50 ng of cDNA at 60 °C for 40 cycles or 62 °C for 35 cycles. The PCR products were separated on a 2% agarose gel and stained with ethidium bromide. Gel images were captured using Quantum ST4 1000/26MX (Fisher Scientific). Specific band were sequenced and were blasted on National Center for biotechnology information (blastn NCBI), and aligned using Clustal Omega program. All the sequences correspond to thrombin (99% homology). Representative gels were selected from at least three independent experiments.

**Measurement of thrombin activity**. Proteolytic activity was measured in Caco-2 cell supernatant samples with BOC-Val-Pro-Arg-amino-4-methylcoumarin hydrochloride (0.5–150 µM, Sigma-Alrich B9395) as substrate in 50 mM Tris, 10 mM CaCl2, 150 mM NaCl, pH = 8.3. Thrombin activity was identified from overall Arg-cleaving enzymes by pre-incubating supernatants with increasing concentrations of the specific thrombin inhibitor lepirudin for 30 min at 37 °C (15.6–1000 pM; Bachem, GmbH), or specific F10 inhibitor Apixaban (0.01 µM, 0.1 µM, 1 µM in 0.1% DMSO vehicle, Selleckchem S1593) for 24-h in serum-free culture medium. Velocity (reaction rate per min) was calculated by the change in fluorescence (excitation: 355 nm, emission: 460 nm), measured over 15 min at 37 °C on a Varioskan Flash microplate reader (Thermo Fisher Scientific). No thrombin activity was detected using this assay in pure FBS nor in heat-inactivated FBS.

**Western blots**. Total protein extract of Caco-2 cells was prepared by using the Nucleospin RNA/Protein kit (Macherey-Nagel, GmbH). Proteins from Caco-2 cell supernatant were precipitated in 15% trichloroacetic acid at 4 °C during 90 min. The pellet was washed twice in cold acetone (−20 °C) and solubilized in 20 µL of protein solving buffer with tris-(2-carboxyethyl)-phosphine hydrochloride (PSB-TCEP; Macherey-Nagel). Samples were then heated at 95 °C for 5 min, clarified by centrifugation at 12,000 × g for 5 min and the solubilized sample was loaded into 4–20% Mini-Protean TGX precast gels (Bio-Rad, GmbH). Human feces from 3 donors (1 g) were suspended in 1 mL PBS buffer, homogenized, centrifuged and supernatants were stored in −20 °C. The mucus was collected in buffer (Tris 40 mM, NaCl 150 mM, EDTA 20 mM, pH 8 and protease inhibitor cocktail, Sigma) by scrapping the colon mucosa of human resection. Lysed samples from human and mouse tissues and feces were diluted in Laemmli buffer 4 × (Biorad), supplemented with 2-mercaptoethanol and heated at 95 °C. Samples were run with at least 20 µg of total protein on Precast Gel 4–15% (Biorad) and transferred to nitrocellulose membrane (Biorad). The membrane was blocked 1 h (PBS, 5% milk and 1% bovine serum albumine) and incubated overnight with anti-thrombin antibody (Santa Cruz sc-16972, 1/200 dilution), anti-Factor 10 (Abcam, Ab79929, 1/200 dilution), or anti-Factor 5 (Abcam, Ab108614, 1/200 dilution), in blocking buffer. Detection was achieved using secondary antibody coupled to horseradish peroxidase (donkey anti-goat IgG, 1/3 000 dilution, Promega V805A), and anti-rabbit (1/3 000, Promega, W401B) during 1 h and a chemiluminescent substrate (ECL from Amersham, Chemidoc XRS, Biorad). Pro-thrombin and active forms of thrombin were used as control (70 and 60 ng of proteins respectively, Sigma-Aldrich). Representative blots were selected from at least three independent experiments.

**N-terminomics/TAILS workflow**. Human microbiota biofilms from 3 different healthy donors have been generated on the Calgary Biofilm Device as describe above. Biofilms were treated with either vehicle (PBS) or recombinant human thrombin (100 mU ml⁻¹, equivalent to 1 nM) for 24 h at 37 °C. Biofilm-associated proteins were extracted in 6 M Urea, 4% SDS buffer, precipitated in trichloroacetic acid (15% final) and were further processed to N-terminomics/TAILS (and shotgun proteomics analysis (Supplementary Figure 10). Briefly, samples were alkylated with iodoacetamide, peptide were then labeled with isotopically heavy [40 mM $^{13}CD_2O$ + 20 mM NaBH₃CN (sodium cyanoborohydride)] or light labels [40 mM light formaldehyde ($CH_2O$) + 20 mM NaBH₃CN]. Samples were then processed for N-terminal enrichment[27,55] and processed to high performance liquid chromatography (HPLC) and mass spectrometry (MS) at the Southern Alberta Mass Spectrometry (SAMS) core facility at the University of Calgary, Canada. Spectral data were matched to peptide sequences in the human UniProt protein database of twelve common bacterial species using the Andromeda algorithm as implemented in the MaxQuant software package v.1.6.0.1, at a peptide-spectrum match false discovery rate (FDR) of < 0.05. The cleavage site specificity was set to semi-ArgC

(free N-terminus), with up to two missed cleavages allowed. Significant outlier cut-off values were determined after log(2) transformation by boxplot-and-whiskers analysis using the BoxPlotR tool[56] (Supplementary Figure 10).

**Statistical analyses**. Graphic representation and statistical analyses were performed using GraphPad Prism (v6, La Jolla, USA). Mann–Whitney's non-parametric t tests were used accordingly after D'Agostino-Pearson normality test. For multiple variables, we used two-way ANOVA followed with Fisher's LSD test. An associated P value less than 5% was considered significant. All central values are means for dot-plots and histograms. Error bars represent standard error of the mean. To represent visual distances in multivariate data set (metagenomic and transcriptomic), we used principal co-ordinate analysis (PCoA) using Past 3 software[57]. Permutational multivariate analysis of variance (PERMANOVA Bonferroni corrected) with Bray-Curtis dissimilarity was used for comparing transcriptomic data set (Past 3). Images for western-blots, RT-PCR and microscopy were obtained from at least 3 independent experiments and/or at least 3 independent human donors.

**Reporting Summary**. Further information on research design is available in the Nature Research Reporting Summary linked to this article.

## Data availability
The sequencing data presented in the manuscript are found in the NCBI database under BioProject PRJNA549826. This project contains the raw 16 S sequencing data deposited in the Sequence Read Archive (SRA) with accession code SRP201956. The raw proteomic data have been deposited in the ProteomeXchange Consortium via the PRIDE partner repository with the data set identifier PXD014315. A reporting summary for this article is available as a Supplementary Information file. The source data used for Fig. 1–6 and Supplementary Figures 1–10 are provided as a Source Data file. Other data supporting the findings of this manuscript are available from the corresponding author upon request.

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

## Acknowledgements

We thank Dr. Claire Racaud-Sultan for fruitful discussions about this project and Corinne Rolland for qRT-PCR primer design. We thank ANEXPLO platforms (UMS 006) animal care facility, histopathology core facility (F. Capilla, and A. Alloy) and the cellular imaging facility (U1043, D. Daviaud and S. Allart). We thank Isabelle Fourquaux for preparing electron microscopy samples (Centre de Microscopie Electronique Appliquée à la Biologie, CMEAB, Faculty of Medicine, Rangueil, Toulouse). Total RNA from primary urothelium was provided by Aude Rubio (IRSD) and human epidermis and HaCat RNA were kindly provided by Dr. Nathalie Jonca (UDEAR CNRS, Toulouse). We thank Dr. S. Brugiroux and Pr. N. Barnich at the University of Clermont-Auvergne for providing us with important bacteria material necessary for the revision of the present work. We thank D. Dubois at the CHU Toulouse for providing human fecal samples. We thank Laurent Brechenmacher at the Southern Alberta Mass Spectrometry (SAMS) core-facility for acquiring the proteomics data. This work was supported by a grant from the European Research Council (ERC- 310973 PIPE) to NV. Tissue collection was originally sponsored by the University Hospital of Toulouse for regulatory and ethic submission, and by a grant from the delegation régionale à la recherche clinique des hôpitaux de Toulouse, through the MICILIP project. The COLIC collection (DC-2015-2443) was also used in the present study. The work used the Aninfimip core facility equipments, supported by Equipex funds from the National "Investments for the future" program (ANR-11-EQPX-0003). JPM was funded by postdoctoral fellowships from the Alberta-Innovate Health Services (AIHS) and from EU's Seventh Framework Programme N° FP7-609398 (AgreenSkills + contract). EFV holds a CRC and is Supported by CIHR MOP# 142773 and a CCC grant in Aid. Equipment acquired thanks to FEDER funding (EU and Occitanie region: Nanorgan project) were used in the present study.

## Author contributions

A.D.S., A.L., C.B., M.Sebbag, D.S., J.P.M., L.G., A.E., NMT identified and characterized the expression and activity of thrombin in epithelial cells, tissues and feces; J.P.M., D.S., L.G., N.M.T., P.R., C.S. characterized the role of thrombin on microbiota in vitro and in vivo; A.D. performed and analyzed N-terminomics/TAILS; J.P.M., A.D.S., C.D. and N.V. analyzed data; D.B. and L.A. provided characterized human clinical samples; J.P.M. and M.Serino analyzed microbiota sequencing; H.J.G. and E.F.V. contributed to germ-free material collection; M.Serino, E.O., M.D.H., A.G.B. and J.L.W. reviewed data, provided intellectual input; J.P.M., C.D. and N.V. wrote the paper; N.V. supervised all aspects of this work. All authors edited and approved the current version of the manuscript.

## Additional information

**Competing interests:** The authors declare no competing interests.

