## [Peer Review File · Nature Communications]

Reviewers' comments:

Reviewer #1 (Remarks to the Author):

In their article Jean-Paul Motta and coworkers present data showing that thrombin is expressed and produced by intestinal epithelial cells. The authors provide evidence that mucosal thrombin has an important function in regulating the commensal microbiota and is needed to proteolyze the biofilm matrix. While the findings are novel and interesting not all conclusions are supported by the results presented.

Major comments:

The data presented are mainly descriptive. For instance, no attempt was made to analyse the composition of the microbiome before and after lepirudin treatment. Also, the molecular mechanisms leading to an activation of thrombin were not analysed and the mode of action of thrombin's ability to reduce total content of matrix-associated proteins was not deciphered. In particular the latter aspect is of importance as the authors state that "thrombin proteolyzed the biofilm matrix". However, no experimental evidence was provided that the proteolytic activity is responsible for the morphological changes of total polysaccharides content of biofilms as seen in upon analysis by scanning electron microscopy.

Considering the impact of the finding it would have been interesting to see whether thrombin expression in germ-free mice can be induced for instance by challenge with LPS or other stimulants. Also experiments are missing showing that thrombin can be used as an antimicrobial treatment to prevent or eradicate biofilm formation caused by for instance antibiotic resistant bacterial.

Minor comments:

Fig. 1A and 2A:

The same set of tissue samples should be used for mRNA transcript analysis. Did the authors made an attempt to detect FV and FX by Western blot or mass spectrometry analysis?

Fig. 1D+F

The different thrombin fragment should be indicated in the figure.

Page 5:

"Since thrombin activity was dosed in supernatants from intestinal epithelial cell line ..."
The meaning of this sentence is not clear and thus the sentence should be re-phrased.

Constitutive thrombin activity preserves mucosal homeostasis:

The authors should indicate whether experiments were performed with conventional or germ-free mice.

Reviewer #2 (Remarks to the Author):

This paper describes a novel observation having potentially important implications for our understanding of the role of thrombin in innate immunity and control of biofilms and bacterial invasion.

As seen from my comments below, one major concern is that presentation of the data needs to be improved. At this level, the authors must see to that all the Methods and materials are described with high precision and clarity.

Another observation that is not well discussed is whether thrombin has direct antibacterial effects, and how this relates to the concentrations of thrombin used in the experiments. As there is evidence that γ -thrombin is present, it is possible that γ -thrombin at high concentrations (near plasma levels) may start to aggregate bacteria or LPS, as shown in a recent paper by Petrlova and colleagues (Petrlova et al., PNAS, 2017). The authors in this ms present evidence that thrombin is bactericidal at higher thrombin concentrations. And it may indeed be so that some of the stained thrombin in the colon crypts may indeed be aggregated, forming amyloids. This could be analysed using thioflavin staining. If the authors do not choose to address this experimentally in the present ms, this section should at least be improved and these possibilities thoroughly discussed.

The authors discuss potential clinical implications and negative side effects, related to the use of anticoagulants. This could have potentially serious implications for the future use of these drugs – given that the observations presented in this ms are valid and clinically relevant. In my opinion, additional evidence should therefore be presented to strengthen the story further. New mouse experiments, where mice are treated systemically with lepirudin and some 1-2 clinically used thrombin inhibitors, followed by evaluation of some 2-3 readouts (histology, bacteria in lymph nodes etc) should therefore be added.

Specific comments:

Page 3. Concerning refs 4, 12 concerning the multiple functions of thrombin some more representative reviews could be selected.

Tissue imaging Fig 1B. It is not mentioned how many patients and samples were analysed for thrombin presence. In the methods part only analyses of animals (blinded and 4 different fields).

Fig 1A. Has it been excluded that the organoids contain inflammatory cells, which could possibly contribute to the thrombin mRNA signal?

Fig 1D. Concerning the identification of PT/T the description of the growth conditions (Methods, cell culture) does not indicate whether serum (eg FBS) was used for the growth for 7 and 24 days, respectively. It is only stated that cells were kept in serum free medium for 24 h. The concern here is whether there could be a carry over effect here, eg that the serum derived T (or PT) is taken up by the cells, and detected during the WB. If so, a control with a FBS sample (that may not be recognised by the Ab, or only detect T since it is serum) should be done. The growth conditions should be indicated better to enable reproductions.

In general, this section should address and exclude the possibility of prothrombin/thrombin contamination from blood, plasma, or the serum addition used in vitro.

Fig. 2 A. The identification of gene transcripts for F5 and F10 does not show that thrombin is produced by the gene products in situ. Thrombin can be autoproteolysed, as well as proteolyzed by many other enzymes, human and bacterial. This should be discussed. The active proteins (X and V) should at least be identified using WB.

Fig. 2 B. This is an interesting observation. In order to add some mechanistic explanation the authors should add data on an in vitro system using previously used epithelial cells stimulated by bacterial supernatants and/or LPS/LTA. If possible a link to TLR/NFkB activation could be analysed (note: correct awkward sentence in the legend).

Fig 3. 10 ug of lepirudin was used for intracolonic administration. How was this dose determined? Was a dose-response study performed? If so it should be mentioned or added in the Suppl data. Has the effect on intracolonic thrombin been verified, ie that the dose is therapeutically active in vivo?

Fig. 3C. There is nothing mentioned about the number of animals used for the histological analyses, or the number of fields analysed (Legends or Methods). Can it be assumed that it is the number of animals in 3A and B? Scoring was done by a "skilled experimenter". How was this performed and according to which criteria? This should be better explained and stated in the ms. Preferably, the authors should redo this and use 2-3 blinded evaluators that use a defined grading scheme.

Another concern is the transcriptomic analysis. Although it may be considered unlikely, lepirudin may show effects on its own. Hence, a control experiment with germ-free animals should be performed just to show that the gene expression is unaffected. If a subset of genes are modulated, this could actually indicate what factors are microbiome-dependent.

Fig. 4. A-C. One representative section is shown and the images are representative of 5 mice. As in 3, there is no information on how the evaluation was performed. As in 3, the authors should re-analyse the sections and use 2-3 blinded evaluators and defined grading scheme. This could be presented as Suppl data.

Fig 5 (and Suppl 5 and 6). These data demonstrate that thrombin shows an effect on biofilm mass and its constituents. What is less clear is whether there is a bactericidal effect or not. Apparently, as seen in Suppl 6, bacteria from donors 3 and 5 are reduced already at 50 mU/ml. As proteolyzed thrombin such as γ -thrombin (not prothrombin or thrombin) has been shown to have aggregation-mediated bactericidal effects at μ M concentrations (Petrlova et al., PNAS, 2017) the reduction could be related to this phenomenon. In order to be able to discuss this possibility we need to know the concentrations, and not only mU, of thrombin used in the experiments in the ms.

From a mechanistic and bacteriological point of view, the authors should complement these analyses using mixtures of bacteria with standard biofilm assays using single bacterial species (eg 4-5 selected), representative for the colon microbial flora.

Fig. 6. As above, detailed specifications on the origin of the materials is lacking. For example, is the skin tissue of human or mouse origin? The procedure for obtaining these materials is not described at all. Is dermis and epidermis included? Another concern is why human keratinocytes were not used.

The experiments show one representative blot (of at least three experiments). The expression level should be better quantified, and compared to the levels from liver cells.

Finally, the language should be improved.

Reviewer #3 (Remarks to the Author):

General comments

The authors show that epithelial thrombin may dictate host-microbial interactions in the gut and perhaps also in other mucosal tissues. The work is highly novel and may have therapeutic implications. It is recommended that the authors not over-interpret their very nice data. A few additional suggestions for revision are offered.

Specific comments

Major

1. To the extent that small numbers of bacteria likely translocate to the liver via the portal circulation in conventionally-raised mice, it is perhaps surprising that hepatic expression of thrombin was unaffected under germ-free conditions. The authors may wish to comment.
2. Discussion para 5. The authors conclude that thrombin is produced "as a physiological response

towards...biofilms". However, they have not shown this specifically, but rather than thrombin production is reduced in germ-free mice. Since the latter animals have numerous other defects, and also lack other components of the microbiota presumably not involved in biofilm formation, the conclusion is over-stated. The authors are also cautioned against claiming primacy for their findings, particularly since many other epithelial products are produced in response to microbial colonization.

Minor

1. There are quite a few language and grammatical errors throughout the manuscript, and particularly missing articles, and mixed tenses within a single sentence. The authors are asked to check the manuscript carefully for these issues.
2. The introduction is a single paragraph that stretches for a page and a half. A logical break would be between the introductory material regarding thrombin, and the discussion of the microbiota and biofilms.
3. Results para 1 (no page numbers). How was it possible to obtain specifically "human crypts supernatants"?
4. Results para 2. What is meant by "Thrombin activity was DOSED in supernatants"?
5. Results para 3. The information about "threshold cycle" for the liver and colon seems to be misplaced, or at least superfluous.
6. The figure legends should be expanded to allow the figures to stand alone from the text. For example, in Figure 3, please add the dose of the thrombin inhibitor. All abbreviations should also be defined, and all statistical tests applied must be stated.
7. In every case where a single image is shown, the authors should state the number of times the experiment was repeated with similar findings. This has been done in some cases but not all.
8. Results para 5. What is meant by "group's complex hulls"?
9. Results para 6. It is a bit of a stretch to refer to mesenteric lymph nodes as "distant organs".

Reviewers' comments and responses from the authors

Reviewer #1 (Remarks to the Author):

In their article Jean-Paul Motta and coworkers present data showing that thrombin is expressed and produced by intestinal epithelial cells. The authors provide evidence that mucosal thrombin has an important function in regulating the commensal microbiota and is needed to proteolyze the biofilm matrix. While the findings are novel and interesting not all conclusions are supported by the results presented. Major comments:

The data presented are mainly descriptive.

Authors: Thrombin is a serine protease produced by hepatocytes in the liver and involved, among other physiological functions, in hemostasis. This is the state of the art regarding the role and origin of thrombin. We thus strongly believe that our results are novel in the field, as this is the first report clearly describing the production of thrombin by intestinal epithelia. Not only we describe an extrahepatic source of such an important protease, but we also unravel a novel role for thrombin in host-microbiota interaction, at least in the gastrointestinal tract. We demonstrated that thrombin acts on bacterial biofilms through its proteolytic capacity of cleaving proteins. We believe that our manuscript has been significantly improved with more mechanistic data generated after your constructive comments. Please find below specific responses and additional data.

For instance, no attempt was made to analyse the composition of the microbiome before and after lepirudin treatment.

Authors: We respectfully remind the reviewer that microbiome composition was analyzed before (day 0) and after (day 10) lepirudin treatment and results were shown in Supplementary Figure 4. These results showed minimal taxonomic changes in microbiome caused by lepirudin treatment. Please find below the figure supporting these observations.

Supp Figure 5. Basal thrombin activity inhibition causes minimal alteration on microbiota taxonomy. C57Bl6 mice were treated daily with either vehicle or lepirudin via an intracolonic route, and euthanized after 10 days (n=5 per

group). Feces were collected at day 0 and at day 10 after lepirudin or vehicle treatment. **A.** 16S Illumina sequencing profiles of bacterial families were determined fecal bacteria from vehicle- and lepirudin treated animals at day 0 (before, B) and day 10 (after, A). **B.** Principal component analysis with filled convex hulls did not reveal a significant shift in overall diversity of fecal microbiota composition between the 4 experimental groups. **C.** Taxonomic cladogram using LEfSe analysis (Linear discriminant analysis effect size) revealed an increased abundance of undetermined Parabacteroides in vehicle-treated group at day 10, and *Alistipes* (Rikenellaceae family) and *Barnesiella* taxa in Lepirudin-treated groups, at day 0 and 10 respectively.

Also, the molecular mechanisms leading to an activation of thrombin were not analysed and the mode of action of thrombin's ability to reduce total content of matrix-associated proteins was not deciphered. In particular the latter aspect is of importance as the authors state that "thrombin proteolyzed the biofilm matrix". However, no experimental evidence was provided that the proteolytic activity is responsible for the morphological changes of total polysaccharides content of biofilms as seen in upon analysis by scanning electron microscopy.

Authors: Thank you for re-emphasizing the importance of thrombin proteolytic activity. We now provide experimental evidence that proteolytic activity is responsible for morphological changes of human complex microbiota biofilms. Indeed, in addition to a condition with "boiled thrombin" that we previously reported, we have performed additional experiments where the effects of thrombin on biofilms were tested in the presence or not of the specific thrombin inhibitor lepirudin. These experiments demonstrated that in the presence of increasing doses of lepirudin, thrombin completely loses its capacity to reduce biofilm biomass (see graph below). These results have been added as the new supplementary Figure 7.

Further, we have investigated the molecular mechanisms by which active thrombin cleaves matrix-associated proteins. Human microbiota biofilms from 3 different healthy donors have been generated on the Calgary Biofilm Device. Biofilms were treated with either vehicle (PBS) or recombinant human thrombin (100 mU/ml, equivalent to 1 nM) for 24 hours at 37°C. Biofilm-

associated proteins were extracted in 6M Urea, 4% SDS buffer, proteins were precipitated in trichloroacetic acid and were further subjected to an N-terminomics/TAILS and shotgun proteomics analysis (Supplementary Figure 10). This unbiased approach has identified multiple substrates of thrombin-related proteolytic activity in addition to identifying the cleavage site position within the biofilm proteins. The list of all significant biofilm-peptides processed by human thrombin were added to the main manuscript as Table 2. This additional set of data hence demonstrate clearly that 1) human thrombin can induce proteolysis of matrix-associated proteins from human microbiota biofilms; 2) novel thrombin substrates within the human microbiota biofilm and the position of the cleavage sites and; 3) human thrombin is able to proteolyze not all (degradation) but specific microbial proteins through targeted processing.

Supplementary Figure 9. N-terminomics/TAILS and shotgun proteomics workflow. A. Schematic representation of N-terminomics/TAILS and shotgun proteomics analysis of three healthy human biofilm samples treated with either vehicle (PBS) or human thrombin (100 mU/ml for 24 hours, equivalent to 1 nM). Peptides were identified by LC-MS/MS and analyzed by MaxQuant. B. Boxplot analysis of the TAILS peptides (n=710). Significant cut-offs were \log_2 0.55 (1.46 fold) for the thrombin-treated group and \log_2 -0.48 (0.51 fold) for the control group.

Table II. Human thrombin cleaves microbial peptides of human multispecies biofilm. Biofilms from three different human healthy donors were treated with vehicle or human thrombin (100 mU/ml) for 24 hours. Biofilm-associated proteins were pooled and processed to N-terminomics TAILS and shotgun proteomics analysis. Spectral data were matched to peptide sequences in the human UniProt protein database of twelve common intestinal bacterial species using the Andromeda algorithm as implemented in the MaxQuant at a peptide-spectrum match false discovery rate of $P < 0.05$. Ratio represents peptide enrichment in thrombin-treated versus vehicle-treated biofilms.

Peptide sequence	Cleavage site	Uniprot ID	Potential protein	Protein function	Potential microbial source	Ratio
SELVESR	122R S123	H1BAZ6;G1V62;A0A1C7HWV8;A0A1C7HWR8;A0A1C7HTV2;A0A1C7HTR9;A0A1C7HS47;A0A1C7HRO4;A0A1C7HRH6;A0A1C7HND4;A0A1C7HLT0;H1B7U0	HMPREF0981_02379	Uncharacterized protein	Erysipelotrichaceae bacterium 6_1_45	64.5
GAEFVARYETDEANVRALD EKMKQFNR	225R G226	A0A1X3COI5	ELS84_0732	ABC superfamily ATP binding cassette transporter, ABC/membrane protein	Enterococcus faecalis (Streptococcus faecalis)	15.3
MLFIEYPTCSTRKAKEYLQ DAGMELEIRHIVEETPTVEE LR	2R M3	G1VQ68	HMPREF9022_02149	Uncharacterized protein	Erysipelotrichaceae bacterium 2_2_44A	12.1
MPDGTGAVSQFEGIPGG KGAVLGGYYPNQEIDFY HR	50R M51	R6UY4	BN746_00221	Uncharacterized protein	Erysipelotrichaceae bacterium CAG:64	3.9
KWYVDATDVPLGR	9R K10	A0A1X3B7M1	rplM	50S ribosomal protein L13	Enterococcus faecalis (Streptococcus faecalis)	3.2
KAGHTALTADIDAQKAIYE	29R K30	A0A2N8HMJ3	CXU19_03050	Uncharacterized	Akkermansia	2.5

ADVTEGASMHSLAAR				protein	muciniphila	
LAESNLR	139R L140	AOA1C7FY3	sigA	RNA polymerase sigma factor SigA	Lachnoclostridium sp. YL32	2.2
NYPGRDAQGAAR	66R N67	AOA2N8HEL8	CXU22_04820	Uncharacterized protein	Akkermansia muciniphila	2.0
GPVASIYQGLMR	126R G127	AOA3A9BVA4	D7W50_04445	MotA/TolQ/ExbB proton channel family protein	Bacteroides caecimuris	1.9
SKDIEVVYLEDLAAEALINEE VR	60R S61	V7ZMF9	arcA	Arginine deiminase	Enterococcus faecalis PF3	1.8
QGISEPMYYQLTGSTEEDLH KQFEGEAETR	323R Q324	V7ZR98	tig	Trigger factor	Enterococcus faecalis PF3	1.8
TDIDLPEQTDKVEALKAS LAE	372R T373	V7ZKR0	metK	S-adenosylmethionine synthase	Enterococcus faecalis PF3	1.8
DLNNYLNKCLR	169R D170	C7V831	EFNG_01499	Aminopeptidase C	Enterococcus faecalis CH188	1.8
SAFTTAAIDLGAHPEYLGAN DIQLGKESVEDTAIVLGSMDGIEFR	61R S62	R3IEF4	WOU_00149	Ornithine carbamoyltransferase	Enterococcus faecalis ATCC 6055	1.8
SLLAEKDFTR	8R S9	R3IEF4	WOU_00149	Ornithine carbamoyltransferase	Enterococcus faecalis ATCC 6055	1.7
ALEEIEAGTVISDPNPEEKR	49R A50	J6R3P9	rpoZ	DNA-directed RNA polymerase subunit omega	Enterococcus faecalis ERV85	1.7
AAVEEGVVAGGVVALIR	404R A405	AOA1C7HIE3	groL	60 kDa chaperonin	Burkholderiales bacterium YL45	1.6
LLFDDIPFLEKAQAEHDFA ELR	36R L37	V7ZMF9	arcA	Arginine deiminase	Enterococcus faecalis PF3	1.5
EFLKERTKGEQTSR	254R E255	AOA1C2G650	BFD03_08580	Uncharacterized protein	Lactobacillus reuteri	1.5
TDAASIEKIAR	235R T236	V7ZMF9	arcA	Arginine deiminase	Enterococcus faecalis PF3	1.5

Considering the impact of the finding it would have been interesting to see whether thrombin expression in germ-free mice can be induced for instance by challenge with LPS or other stimulants.

Authors: We agree with the reviewer that it is interesting to see whether thrombin expression in germ-free mice can be restored by bacterial motives. We already had several *in vitro* data that we want to share here with the reviewers, and we have also performed additional experiments that have been added to the manuscript.

First, we observed *in vitro*, that thrombin expression is already turned on in intestinal epithelial cell lines, but we tried to investigate whether the expression and thrombin activity could be modified by LPS exposure. When exposed to TLR-4 agonist such as E.coli K12 LPS (5 µg/ml), thrombin-activity released by Caco-2 cell or expression were not modified (see results below).

We also investigated the effects of different TLR ligands; agonists of TLR2 with *heat-killed Listeria monocytogenes* (HKLM, 10^8 cells/ml), TLR3 with (Poly(I:C high-molecular weight HMW (50 µg/ml), TLR 6/2 with fibroblast-stimulating lipopeptide (FSL-1, 100 ng/ml) and TLR8 with single strand RNA 40 (5 µg/ml), at respectively) still *in vitro* in Caco-2 cell cultures and could not observe further regulation of thrombin expression in intestinal epithelial

cells. Altogether, we believe that microbial colonization is necessary to induce good level of thrombin expression in intestinal epithelial cells, but that once this expression is turned on, further regulation by bacterial motives on TLRs might not be that important.

In vivo, the question of the induction of thrombin expression by bacterial motives remains open, and we have tried to address it by new experiments.

Although the gastrointestinal tract of germfree animals can be considered sterile, it is still permanently exposed to self-antigens, ingested food antigens and microbial residues in sterile food or beddings, such as endotoxin (Kim et al 2016). Interestingly, bacterial residues in sterile chow of GF mice have been associated with expansion of B and T cells in the gut associated lymphoid tissue (Hrcir et al 2008). Endotoxin contamination has been reported in ST1 and R03 diets, both grain-based diets which are routinely used after irradiation to feed GF animals (Schwarzer et al 2017). **These data clearly suggest that the contamination of sterile food with bacterial residues such as LPS and other endotoxins is expected in GF animals.** For that reason, we choose to challenge germ-free mice not only with LPS, but instead to investigate whether thrombin expression could be recapitulated in germ-free mice recolonized with pathogen-free standard microbiota. We recolonized 5 germfree animals (2 females, 3 males) with adult SPF animals cecal content by oral gavage, and harvested tissues (liver and colon mucosa) after three weeks. This experiment demonstrated that although GF mice have a dramatic reduction of F2 mRNA transcription in colon mucosa, a recolonization with SPF microbiota is sufficient to recover a basal transcription of epithelial thrombin. Epithelial thrombin is thus likely to be regulated by local bacterial metabolites or with bacterial contact. This is in clear contrast with the regulation of F2 mRNA in the liver, which seems independent of gut microbial colonization. These results have been added to the new Figure 2 panel D.

- Kim KS, Hong S-W, Han D, Yi J, Jung J, Yang B-G, et al. Dietary antigens limit mucosal immunity by inducing regulatory T cells in the small intestine. *Science* (80-). 2016;351: 858–863.

- Hrnčir T, Stepankova R, Kozakova H, Hudcovic T, Tlaskalova-Hogenova H. Gut microbiota and lipopolysaccharide content of the diet influence development of regulatory T cells: studies in germ-free mice. BMC Immunol. 2008;9: 65 10.1186/1471-2172-9-65

- Schwarzer M, Srutkova D, Hermanova P, Leulier F, Kozakova H, Schabussova I. Diet matters: endotoxin in the diet impacts the level of allergic sensitization in germ-free mice. PLoS ONE (2017) 12:e0167786. 10.1371

Germfree mice (GF, n=5), germfree recolonized with specific-pathogen free (SPF) cecal content (GF>SPF, n=5) and conventionally-bred mice (CONV, n=8) were sacrificed at 10-14 weeks old. Relative expression of *F2* mRNA transcript (thrombin) was unchanged in germ-free mouse liver compared to conventionally-bred mouse liver and GF>SPF. In germ-free mouse, *F2* mRNA was significantly reduced in colon mucosa compared to conventionally-bred mouse colon mucosa. GF recolonized with SPF microbiota had identical transcription of epithelial thrombin compared to conventional mice. One-way ANOVA with Fisher's LSD test ** P < 0.01 versus GF group.

[Redacted]

[Redacted]

Minor comments:

Fig. 1A and 2A:

The same set of tissue samples should be used for mRNA transcript analysis. Did the authors made an attempt to detect F5 and F10 by Western blot or mass spectrometry analysis?

Authors: Indeed, the same set of samples were used in Fig. 1A and 2A to detect FII (Fig. 1A), F5 and F10 (Fig. 2A) in Caco-2 cells and HT-29 cells. The same mRNA extraction and the same house-keeping gene were used. This is now specified in Figure 2 legend.

To answer the reviewer's comment, we have investigated the presence of F5 and F10 proteins in cell lysates of Caco-2.

Factor 10 (as an inactive zymogen) is a disulfide-linked heterodimer previously demonstrated to be produced only by hepatocytes. It consists of two chains resulting from a 74-76 kDa single chain precursor. This creates a 55-57 kDa heavy chain, and a 17-18 kDa light chain. Generation of active F10 (F10a) results in the cleavage of the heavy chain, leading to the generation of a 45-46 kDa active chain. Factor 5 (F5) circulates in blood as a single chain protein (330 kDa). Following proteolytic activation by thrombin, activated factor 5 (F5a) serves as the cofactor for factor 10a in the prothrombinase complex that cleaves prothrombin to thrombin. Factor 5a is composed of a heavy chain (94 kDa) non-covalently associated to a light chain (74 kDa).

Western blots revealed the presence of Factor 10 (inactive, 74 kDa and active form 54 kDa, left blot) as well as Factor 5 (inactive, 300 kDa ; active heavy chain 94 kDa and active light chain 74 kDa, right blot) in Caco-2 cell protein extract (left blot) .

This analysis was included in revised Figure 2 panel B.

B.

Finally, we have investigated whether thrombin activity released by intestinal epithelial cells was dependent on the activation of F10. In the presence of the F10 inhibitor Apixaban (0.01 μ M, 0.1 μ M and 1 μ M), thrombin activity produced by intestinal epithelial cells was dose-dependently and significantly inhibited. This further suggests that F10 is present in intestinal epithelial cell cultures and involved in thrombin activation. These results have been added to Figure 2.

Epithelial thrombin activity produced by Caco-2 is dose-dependently inhibited by the presence of the F10 inhibitor Apixaban during 24 hours (0.01 μ M, 0.1 μ M and 1 μ M). Activity assay was reproduced in 3 independent experiments, with 9 replicates per concentration.

Fig. 1D+F The different thrombin fragment should be indicated in the figure.

Authors: We have modified Figure 1 panel 1D and 1E as well as Supplementary Figure 1.

Page 5:

“Since thrombin activity was dosed in supernatants from intestinal epithelial cell line ...”

The meaning of this sentence is not clear and thus the sentence should be re-phrased.

Authors: The sentence has been re-phrased for better clarity.

Constitutive thrombin activity preserves mucosal homeostasis:

The authors should indicate whether experiments were performed with conventional or germ-free mice.

Authors: Except in Figure 2, where we investigated expression of thrombin expression in germfree mice from McMaster Germ-Free Facility, all other animal studies have been performed using SPF animals (specific pathogen free) purchased from Janvier laboratories (France) and bred in ANEXPLO platforms (UMS 006, Toulouse) animal care facility.

Reviewer #2 (Remarks to the Author):

This paper describes a novel observation having potentially important implications for our understanding of the role of thrombin in innate immunity and control of biofilms and bacterial invasion.

Authors: Thank you for this positive comment. Please find below specific response and novel data we provide after your inquiries, and that we believe have significantly improved our study.

As seen from my comments below, one major concern is that presentation of the data needs to be improved. At this level, the authors must see to that all the Methods and materials are described with high precision and clarity.

Authors: Thank you for this comment, we have revised Methods and Material section to describe experiments that have been performed with the highest precision and clarity.

Another observation that is not well discussed is whether thrombin has direct antibacterial effects, and how this relates to the concentrations of thrombin used in the experiments. As there is evidence that γ -thrombin is present, it is possible that γ -thrombin at high concentrations (near plasma levels) may start to aggregate bacteria or LPS, as shown in a recent paper by Petrlova and colleagues (Petrlova et al., PNAS, 2017). The authors in this ms present evidence that thrombin is bactericidal at higher thrombin concentrations. And it may indeed be so that some of the stained thrombin in the colon crypts may indeed be aggregated, forming amyloids. This could be analysed using thioflavin staining. If the authors do not choose to address this experimentally in the present ms, this section should at least be improved and these possibilities thoroughly discussed.

Authors: Thanks to the reviewer's comment, we have toned down our statement that thrombin does not have direct antibacterial effects. Indeed, we only have tested antibacterial effects of thrombin in specific conditions of doses and time of incubation. Therefore, we have re-worded the manuscript in agreement with this reviewer's suggestion. We underscore that the lack of antibacterial effect of thrombin was limited to our specific experimental conditions, and that thrombin might indeed have antibacterial effects *in vivo*, especially if we consider previously published data from Petrlova et al. This paper is now cited in the manuscript.

In addition, as suggested by the reviewer, we have performed thioflavin staining of human colonic tissue samples from three different healthy donors. We observed that thioflavin staining did not co-localize with thrombin staining in the colons, indicating that thrombin might not form or associate with amyloid protein aggregates, at least in homeostatic conditions. It could be different in infectious conditions such as the one described in the Petrlova et al. paper, but this particular condition would be out of the scope of the present paper. These results have been added as supplementary Figure 2.

Supplementary Figure 2. Epithelial thrombin do not form, or associated with, amyloid fibrils protein aggregates in healthy colon mucosa. Immunostaining of human thrombin (F2, in red) do not colocalize with amyloid-like protein

aggregates (Thioflavin T staining, in yellow) in human colon biopsy from three different donors. Host cell nuclei were stained with 4',6-diamidino-2-phenylindole or DAPI. Scale bar corresponds to 50 μ m.

The authors discuss potential clinical implications and negative side effects, related to the use of anticoagulants. This could have potentially serious implications for the future use of these drugs – given that the observations presented in this ms are valid and clinically relevant. In my opinion, additional evidence should therefore be presented to strengthen the story further. New mouse experiments, where mice are treated systemically with lepirudin and some 1-2 clinically used thrombin inhibitors, followed by evaluation of some 2-3 readouts (histology, bacteria in lymph nodes etc) should therefore be added.

Authors:

We agree with the reviewer that the potential clinical implications on the GI tract of the use of anticoagulant therapies are major, and that experiments investigating the GI side effects of such therapies would strengthen the paper presented here. In agreement with this comment, we performed experiments to assess this potential. We chose not to include the results of these experiments in the main manuscript, but instead to illustrate them as supplemental data, in order to keep our message focused on the presence of thrombin and its effects on microbial biofilms in the distal colon.

In order to be clinically relevant, we choose to treat groups of mice with 2 different anticoagulant treatments that are classically used in patients as oral treatments. One group of mice (n=12) was treated orally with the direct thrombin inhibitor dabigatran etexilate or brand name Pradaxa© (50 mg/kg/day in mouse corresponding to human dose of ~4 mg/kg/day). Another group (n=7) was given warfarin or brand name Coumarin© in drinking water *ad libitum* (10 mg/L). Final dose based on water consumption dose intake was 2mg/kg/day, which is equivalent to a human dose of 10 mg daily recommended for prevention of thromboembolism (RxList.com). Warfarin inhibits epoxide reductase (specifically the VKORC1 subunit), thereby diminishing availability of vitamin-K in tissues, and subsequently of the glutamyl carboxylase enzyme. When this occurs, the coagulation factors are no longer carboxylated, and are thus biologically inactive. Warfarin thus impairs synthesis of active coagulation factors II (thrombin), VII, IX and X, as well as the regulatory factors protein C, protein S, and protein Z. Finally, a control group of 12 mice was orally treated with vehicle (water).

As requested by the reviewer, we recorded daily fecal score based on consistency (score 0 for normal feces, 1 for soft feces, 2 for diarrhea) and presence of blood (Hemocult tests, score 0 for negative, score 1 for positive, score 2 for gross bleeding). We also investigated potential macroscopic lesions in the stomach (purpura/petechia, tissue hematoma and edema), as well as bacterial translocation in mesenteric lymph nodes (for presence of aerobic and anaerobic bacteria). Results are presented in the graph below and were included as Supplementary Figure 9. Taken together, these data show that oral treatment with anticoagulant targeting directly or indirectly the activation of thrombin induce gastro-intestinal damage, leading to bacterial translocation.

Supp Figure 9. Gastrointestinal damage induced by oral anticoagulant treatments Dabigatran (thrombin inhibitor) and warfarin (vitamin K antagonist). C57Bl/6 mice were treated daily with either vehicle (n=12), warfarin (n=7, 10 mg/L in drinking water *ad libitum*), or dabigatran etexilate (50 mg/kg/day oral gavage, n=12) for 7 days and euthanized for further analysis (A to D). A. Change in body weight and B. fecal score (consistency and presence of blood) were recorded daily. C. Macroscopic damage score (purpura/petechia, tissue hematoma and edema) in both warfarin and dabigatran etexilate-treated group were significantly greater than the scores measured in vehicle controls. D. Bacterial translocation of aerobes (warfarin) and anaerobes (warfarin and dabigatran-etexilate), into mesenteric lymph nodes (MLNs) was significantly greater in animals treated with oral anticoagulants compared to vehicle-treated animals. One-way ANOVA test with Fisher's LSD * $P < 0.05$ versus CTR group.

To summarize, these novel *in vivo* data are consistent with our previous findings demonstrating that thrombin activity in mucosa is protective, not only in the distal colon but also in the upper GI-tract. As suggested by the reviewer, these observations should have serious implications for one of the most commonly prescribed drugs in human (ie. Oral anticoagulants). Nevertheless, in the upper GI-tract, whether epithelial thrombin plays a role in local microbiota organization, as it does in distal colon, will require thorough and dedicated investigations in the future.

Specific comments:

Page 3. Concerning refs 4, 12 concerning the multiple functions of thrombin some more representative reviews could be selected.

Authors: We agree with the referee that references 4 and 12, although discussing thrombin specifically, are reviews more focused on the role of proteases in general with a particular focus on PAR receptor activation.

Nevertheless, we have included, and kept, 2 important reviews describing the multiple effects and functions of thrombin.

- Siller-Matula JM, Schwameis M, Blann A, Mannhalter C, Jilma B. Thrombin as a multi-functional enzyme. Focus on *in vitro* and *in vivo* effects. *Thromb Haemost* 106, 1020-1033 (2011).
- Huntington JA. Thrombin plasticity. *Biochim Biophys Acta* 1824, 246-252 (2012).

Tissue imaging Fig 1B. It is not mentioned how many patients and samples were analysed for thrombin presence. In the methods part only analyses of animals (blinded and 4 different fields).

Authors: Panel in Figure 1B is a representative image of thrombin immunostaining that has been performed using biopsies from three different human healthy donors (colon cancer screening and healthy endoscopic mucosa). Immunostaining for F2 has been performed at least 3 times independently, leading to the same observations (presence of thrombin in the colon epithelium, as well as in the colon lumen area). As rightly observed by the reviewer's, this information is now included clearly in the figure legend.

Fig 1A. Has it been excluded that the organoids contain inflammatory cells, which could possibly contribute to the thrombin mRNA signal?

Authors: Yes, we have excluded that our regular organoid cultures contain inflammatory cells. We have performed CD45+ cell staining, which was completely negative in our organoid cultures, but positive in lamina propria isolated cells. This has been specified in the Method section.

Fig 1D. Concerning the identification of PT/T the description of the growth conditions (Methods, cell culture) does not indicate whether serum (eg FBS) was used for the growth for 7 and 24 days, respectively. It is only stated that cells were kept in serum free medium for 24 h. The concern here is whether there could be a carry over effect here, eg that the serum derived T (or PT) is taken up by the cells, and detected during the WB. If so, a control with a FBS sample (that may not be recognised by the Ab, or only detect T since it is serum) should be done. The growth conditions should be indicated better to enable reproductions. In general, this section should address and exclude the possibility of prothrombin/thrombin contamination from blood, plasma, or the serum addition used in vitro.

Authors: We agree with the reviewer that it is very important to consider the possibility of a prothrombin/Thrombin contamination from serum, and indeed we have performed a number of controls that we did not necessarily presented in the manuscript, but that we want to discuss here. Fetal Bovine Serum (FBS) was used at the beginning of the intestinal epithelial cell cultures, but was removed 24h before harvesting the cells and cell supernatants. This was already specified in the methods, and is now more clearly stated in the legend of Figure 1. We are confident that no thrombin carry over from serum could be detected in intestinal epithelial cell line cultures for several reasons:

1/ Human thrombin mRNA is detected in cultured human intestinal epithelial cells, meaning that these cells are capable of producing thrombin. mRNA of human thrombin could not come from FBS.

2/ We have measured thrombin activity in FBS used for initial cell cultures and detected no thrombin activity, meaning that if some thrombin is present in FBS, it is not active and could not account for the thrombin activity released by intestinal epithelial cells grown for 24h in serum-free conditions.

3/ We have used Activity-Based Probes to harvest active proteases from human intestinal epithelial cell cultures (after 24h of serum-free cultures). We have then run mass spectrometry analysis on the harvested proteases and detected peptides corresponding to human thrombin. Precisely, the mass spec analysis identified 21 peptides with similarity to human thrombin. Among them, 8 peptides showed a perfect alignment to human thrombin (no internal gaps within the alignment). Among these 8 peptides, 5 of them had 100% identity with human thrombin and mismatched bovine thrombin, one peptide matched only human thrombin, and only 2 peptides were conserved both in human and bovine thrombin. No peptide presented higher identity to bovine thrombin than to human thrombin, and no peptide aligned exclusively to bovine thrombin. The algorithm used for this analysis was blastp, the data base was reference proteins (refseq-protein) and the organisms considered were homo sapiens (taxid:9606) and Bos Taurus (taxid:9913). The query entry was prothrombin.

Taken together, these data constitute a formal proof that the human thrombin protein is present in supernatants from intestinal epithelial cells in culture.

As requested by the reviewer, we have revised the method section to add details on the way experiments were performed in order to facilitate reproducibility:

Cell culture. Human intestinal epithelial cell lines (Caco-2, HT-29, SW480; ATCC, USA) were grown in DMEM high glucose GlutaMAX supplemented with 1x non-essential aminoacids, 1x penicillin/streptomycin and 10% FBS (Gibco). Briefly, 3×10^5 IECs were plated on flat-bottom 6-well plates and grown for 7 days at 37°C 5% CO₂ with culture medium replacement three times a week. Additionally, 1×10^5 Caco2 cells were plated on polycarbonate 12-transwells plates and grown for 21 days, as described above. For harvesting of cellular supernatants, cells were washed twice with PBS Ca²⁺/Mg²⁺-free and kept for 24 h in the cell culture medium described above, but without FBS.

Fig. 2 A. The identification of gene transcripts for F5 and F10 does not show that thrombin is produced by the gene products in situ. Thrombin can be autoproteolysed, as well as proteolyzed by many other enzymes, human and bacterial. This should be discussed. The active proteins (X and V) should at least be identified using WB.

Authors: We agree with the reviewer that the generation of active thrombin can be achieved in many different ways, and we have added this notion in the discussion section of the manuscript. To answer the reviewer's comment, we have investigated the presence of F5 and F10 proteins in cell lysates of Caco-2.

Factor 10 (as an inactive zymogen) is a disulfide-linked heterodimer previously demonstrated to be produced only by hepatocytes. It consists of two chains resulting from a 74-76 kDa single chain precursor. This creates a 55-57 kDa heavy chain, and a 17-18 kDa light chain. Generation of active F10 (F10a) results in the cleavage of the heavy chain, leading to the generation of a 45-46 kDa active chain. Factor 5 (F5) circulates in blood as a single chain protein (330 kDa). Following proteolytic activation by thrombin, activated factor 5 (F5a) serves as the cofactor for factor 10a in the prothrombinase complex that cleaves prothrombin to thrombin. Factor 5a is composed of a heavy chain (94 kDa) non-covalently associated to a light chain (74 kDa).

Western blots revealed the presence of Factor 10 (inactive, 74 kDa and active form 54 kDa, left blot) as well as Factor 5 (inactive, 300 kDa ; active heavy chain 94 kDa and active light chain 74 kDa, right blot) in Caco-2 cell protein extract (left blot) .

This analysis was included in revised Figure 2 panel B.

Finally, we have investigated whether thrombin activity released by intestinal epithelial cells was dependent on the activation of F10. In the presence of the F10 inhibitor Apixaban (0.01 μ M, 0.1 μ M and 1 μ M), thrombin activity produced by intestinal epithelial cells was dose-dependently

significantly inhibited, thereby further suggesting that F10 is present in intestinal epithelial cell cultures and involved in thrombin activation. These results have been added to Figure 2.

Epithelial thrombin activity produced by Caco-2 is dose-dependently inhibited by the presence of the F10 inhibitor Apixaban during 24 hours (0.01 μ M, 0.1 μ M and 1 μ M). Activity assay was reproduced in 3 independent experiments, with 9 replicates per concentration.

Fig. 2 B. This is an interesting observation. In order to add some mechanistic explanation the authors should add data on an *in vitro* system using previously used epithelial cells stimulated by bacterial supernatants and/or LPS/LTA. If possible a link to TLR/NF κ B activation could be analysed (note: correct awkward sentence in the legend).

Authors: We agree with the reviewer that a mechanistic explanation would strengthen this observation.

We observed *in vitro*, that thrombin expression is already turned on in intestinal epithelial cell lines and that addition of a TLR-4 agonist such as E.coli K12 LPS (5 μ g/ml) did not increase thrombin-activity released by Caco-2 cell (see results below).

We also investigated the effects of different TLR ligands; agonists of TLR2 with *heat-killed Listeria monocytogenes* (HKLM, 10^8 cells/ml), TLR3 with (Poly(I:C high-molecular

weight HMW (50 $\mu\text{g/ml}$), TLR 6/2 with fibroblast-stimulating lipopeptide (FSL-1, 100 ng/ml) and TLR8 with single strand RNA 40 (5 $\mu\text{g/ml}$), at respectively) still *in vitro* in Caco-2 cell cultures and could not observe further regulation of thrombin expression in intestinal epithelial cells. Altogether, we believe that microbial colonization is necessary to induce good level of thrombin expression in intestinal epithelial cells, but that once this expression is turned on, further regulation by bacterial motives on TLRs might not be that important.

To further understand epithelial F2 regulation by microbial triggers, we choose to challenge germ-free mice not with LPS, but instead to investigate whether thrombin expression could be recapitulated in germ-free mice recolonized with pathogen-free standard microbiota. We recolonized 5 germfree animals (2 females, 3 males) with adult SPF animals cecal content by oral gavage, and harvested tissues (liver and colon mucosa) after three weeks. This experiment demonstrated that although GF mice have a dramatic reduction of F2 mRNA transcription in colon mucosa, recolonization with SPF microbiota is sufficient to recover a basal transcription of epithelial thrombin. Epithelial thrombin is thus likely to be regulated by local bacterial metabolites or by bacterial contact. This is in clear contrast with the regulation of F2 mRNA in the liver, which appears independent of the gut microbial colonization. These results have been added to the new Figure 2 panel D.

Germfree mice (GF, n=5), germfree recolonized with specific-pathogen free (SPF) cecal content (GF>SPF, n=5) and conventionally bred mice (CONV, n=8) were sacrificed at 10-14 weeks old. Relative expression of *F2* mRNA transcript (thrombin) was unchanged in germ-free mouse liver compared to conventionally bred mouse liver and GF>SPF. In germ-free mouse, *F2* mRNA was significantly reduced in colon mucosa compared to conventionally bred mouse colon mucosa. GF recolonized with SPF microbiota had identical transcription of epithelial thrombin compared to conventional mice. One-way ANOVA with Fisher's LSD test ** P < 0.01 versus GF group.

The odd sentence in the legend of Figure 2 was corrected.

Fig 3. 10 ug of lepirudin was used for intracolonic administration. How was this dose determined? Was a dose-response study performed? If so it should be mentioned or added in the Suppl data. Has the effect on intracolonic thrombin been verified, ie that the dose is therapeutically active in vivo?

Authors:

In our experiment, we administered a dose of 10 µg/day via intracolonic route for 10 days. To determine this dose, we reasoned that:

1/ Human dose for lepirudin treatment was 0.1 to 0.4 mg/kg intravenously. This leads to a mouse equivalent dose of 1 to 4 mg/kg (10 to 40 µg).

2/ Previous studies reported that Lepirudin has been used in mice at a dose of 1.5 mg/kg (30µg) in a single dose administered intravenously (Lauer et al 2011).

3/ One molecule of lepirudin binds to one molecule of thrombin and thereby blocks the thrombogenic activity of thrombin in equimolar ratio. Considering that 1 NIH U/ml of active thrombin correspond to 0.3 µg of protein (Sigma Aldrich), that thrombin activity released in 1 cm² by intestinal mucosa was 1 to 5 mU/ml, and the mouse GI tract surface = 1.4 m² (Casteleyn et al 2010), this led to a total amount of 4 to 20 µg of thrombin released at the mucosal surface.

4/ Lepirudin does not cross the blood-brain barrier (Markwardt et al 1988) and or the placental barrier (Lindhoff-Last et al 2000). It is thought to be metabolized by release of amino acids via catabolic hydrolysis. For this reason, and in contrast with other direct anticoagulants (dabigatran) or vitamin K antagonists (warfarin), lepirudin (which was discontinued for human use since 2012) was prescribed via intravenous route. We thus believe that 10 µmg of lepirudin when administered intracolonicly inhibited most of the epithelial thrombin released in the lumen

- Markwardt F, Fink G, Kaiser B, et al. Pharmacological survey of recombinant hirudin. *Pharmazie*. (1988);43:202–7
- Lindhoff-Last E, Willeke A, Thalhammer C, et al. Hirudin treatment in a breastfeeding woman. *Lancet*. (2000);355:467–8
- A. Lauer, F. A. Cianchetti, E. M. Van Cott et al., Anticoagulation with the oral direct thrombin inhibitor dabigatran does not enlarge hematoma volume in experimental intracerebral hemorrhage, *Circulation*, (2011); 124:1654–62
- Casteleyn C., Rekecki A., Van der Aa A., Simoens P., Van den Broeck W. Surface area assessment of the murine intestinal tract as a prerequisite for oral dose translation from mouse to man. *Lab. Anim.* (2010); 44: 176-183

Fig. 3C. There is nothing mentioned about the number of animals used for the histological analyses, or the number of fields analysed (Legends or Methods). Can it be assumed that it is the number of animals in 3A and B? Scoring was done by a "skilled experimenter". How was this performed and according to which criteria? This should be better explained and stated in the ms. Preferably, the authors should redo this and use 2-3 blinded evaluators that use a defined grading scheme.

Authors: The number of animals used in Figure 3 is the same for Figure 3A, B, C and D, this is now made clearer in the Figure legend.

Histology scoring was performed according to a previously published scoring system which was cited but this citation was not explicitly explained. We have now corrected this sentence in the methods section. We have also added the number of fields analyzed: 5 per tissue section.

The histology scoring has been redone blindly of the treatments by another evaluator. Results obtained did not change initial observations.

Histological changes	Microscopic Damage Score (Chinen et al 2011 Nat Commun)			
	0	1	2	3
Loss of goblet cells	None	Mild/focal	Numerous and diffuse	Numerous and diffuse
Crypt abscesses	None	Few (1 in several sections)	A few (1-4 per section)	Many (≥ 5 per section)
Hyperaemia in the mucosa	None/not evident	Mild	Diffuse	Diffuse
Cellular infiltration in the lamina propria	None	A few	Moderate/focal	Numerous and diffuse
Elongation of colonic mucosa	Normal ($\leq 100\%$)	Mild (101–150%)	Evident ($\geq 150\%$)	Evident ($\geq 150\%$)
Epithelial erosion	None	Mild/focal	Evident/diffuse	Evident/diffuse

Another concern is the transcriptomic analysis. Although it may be considered unlikely, lepirudin may show effects on its own. Hence, a control experiment with germ-free animals should be performed just to show that the gene expression is unaffected. If a subset of genes are modulated, this could actually indicate what factors are microbiome-dependent.

Authors: We agree with the reviewer that this is a point of potential concern. However, we face a feasibility issue on that question. Repeated (daily for 10 days) administrations in germ-free mice is extremely complicated to perform if we want to keep the germ-free status of the animals. Indeed, it requires specific isolators and equipment that our collaborators at McMaster Germ-free core facility could not provide. Alternatively, we performed *in vitro* experiments with biofilm cultures, demonstrating that Lepirudin alone had no effect on the biofilms (new supplementary Figure 7). This at least addresses the point from the microbiota point of view.

Lepirudin does not reduce biomass of human multispecies. Multispecies anaerobic biofilms were generated from one human healthy donor. Mature biofilms were then exposed to various concentrations of lepirudin (thrombin irreversible inhibitor). At concentration used, lepirudin alone has no effect on total biofilm biomass.

Fig. 4. A-C. One representative section is showed and the images are representative of 5 mice. As in 3, there is no information on how the evaluation was performed. As in 3, the authors should re-analyse the sections and use 2-3 blinded evaluators and defined grading scheme. This could be presented as Suppl data.

Authors: We agree with the reviewer that a grading scheme should be defined for this specific analysis, but to the best of our knowledge, nothing related to microbiota spatial alterations have yet been described in the literature. We therefore took this opportunity to propose a scoring system and to reanalyze our data according to that novel scoring system. We propose to establish a “biofilm damage score” which is designed to consider any spatial alterations that could be observed in microbiota ranging from a normal microbiota in healthy animals to a completely altered microbiota as observed in severe colitis, see references Motta et al 2015 IBD and Motta et al 2018 IBD.

We have blindly reanalyzed microscopic fields (13 in vehicle, 22 in Lepirudin) and obtained an average biofilm damage score for each group of mice, revealing a higher damage score in lepirudin-treated animals. We believe such scoring system illustrates more objectively the spatial alterations observed in microbiota after lepirudin treatment.

We have decided to include this novel scoring data in the revised Figure 4 and Table 1.

Table I (Biofilm damage score).

	Biofilm Damage Score			
	0	1	2	3
Inner mucus layer invasion	Rare	Few	Numerous	Numerous and dense colonies
Biofilm distance to epithelia	>20µm	>10µm		Numerous contact
Biofilm density	High	Mild	Scattered	Mostly planktonic
Bacterial translocation into lamina propria	None	Few	Numerous	Dense colonies
Immune cell transmigration into the biofilm	None	Scattered	Dense	Dense and extracellular DNA
Bacteria morphology	More than 5 morphotypes	<4 morphotypes	<3 morphotypes	Predominance of one morphotype

Figure 4. Suppression of basal thrombin activity modifies spatial organization of microbiota biofilm. C57Bl/6 mice were treated daily with either vehicle (n=5) or lepirudin (n=10, 10 µg) via intracolonic route, euthanized after 10 days. Distal colon sections were Carnoy’s fixed. All bacterial cells were labeled with the universal probe Eub338 for fluorescent *in situ* hybridization. Wheat germ agglutinin was used to stain the polysaccharides-rich mucus layer on these samples. Based on Table I (Biofilm damage score), abnormal alterations to gut microbiota biofilm organization was blindly recorded (13 fields in vehicle and 22 fields in lepirudin) and demonstrated an increased damage score in animals treated with lepirudin compared to vehicle control groups. Unpaired Mann-Whitney test * P < 0.001

Fig 5 (and Suppl 5 and 6). These data demonstrate that thrombin shows an effect on biofilm mass and its constituents. What is less clear is whether there is a bactericidal effect or not. Apparently, as seen in Suppl 6, bacteria from donors 3 and 5 are reduced already at 50 mU/ml. As proteolyzed thrombin such as γ -thrombin (not prothrombin or thrombin) has been shown to have aggregation-mediated bactericidal effects at μ M concentrations (Petrlova et al., PNAS, 2017) the reduction could be related to this phenomenon. In order to be able to discuss this possibility we need to know the concentrations, and not only mU, of thrombin used in the experiments in the ms.

Authors:

As mentioned above, we have toned down the statement that epithelial thrombin is not bactericidal. We have performed thioflavin staining demonstrating that unlike what is shown in Petrlova et al., in healthy colons thrombin does not co-localize with aggregates. Mention of that paper has been made in the manuscript and thioflavin staining has been added (Supp Figure 2).

In addition, we have added to all the legends where thrombin was used, the equivalent dose in molarity of thrombin, so readers could have a sense of molecule numbers. However, as enzymologists, and because we discuss our data in view of proteolytic effects of thrombin, we believe it is important to reason also in enzyme units and to keep this information in our figures.

From a mechanistic and bacteriological point of view, the authors should complement these analyses using mixtures of bacteria with standard biofilm assays using single bacterial species (eg 4-5 selected), representative for the colon microbial flora.

Authors: We thank the reviewer for this comment. We have complemented our study using a mixture of intestinal bacteria grown anaerobically as a biofilm phenotype onto our Calgary Biofilm Device (*Bacteroides sp.*, *Clostridium innocuum*, *Lactobacillus reuteri* and *Enterococcus faecalis*; gift from collaborators Dr. S. Brugiroux and Pr. N. Barnich, University of Clermont-Auvergne). Additionally, we cultured each strain independently and exposed mature 48 hours biofilms to increasing concentrations of human active thrombin (125, 250, 500 and 1000 mU/ml) for 24 hours. Biofilm biomass was measured and revealed that mix of 4 strains, as well as *Clostridium innocuum* and *Enterococcus faecalis* biofilms were all dose-dependently affected by human thrombin. Biofilms of *Bacteroides* and *Lactobacillus reuteri* did not respond to thrombin, suggesting that some but not all bacterial matrix components are proteolyzed by thrombin. These results are consistent with our data using complex multispecies biofilms from human donors.

Fig. 6. As above, detailed specifications on the origin of the materials is lacking. For example, is the skin tissue of human or mouse origin? The procedure for obtaining these materials is not described at all. Is dermis and epidermis included? Another concern is why human keratinocytes were not used.

Authors: We respectfully remind to the reviewer that we have specified in Figure legends as well as in the Figure itself whether tissue came from human (panel A) or mouse (panel B).

In the present paper, we have indeed not investigated *F2* mRNA expression in human keratinocytes. However, we used cDNA from healthy human epidermis tissue (Figure 6 panel A) which consists of 90% of keratinocytes, but also melanocytes, Langerhans cells as well as potential immune cells. These cDNA came from healthy human skin processing and were kindly provided by our colleagues at the UDEAR CNRS as acknowledged in the manuscript.

Nevertheless, additional data presented below demonstrate the presence of *F2* mRNA in cDNA extracted from human epidermis (Figure 5, figure below lane 1), as well as in *ex vivo* cultures of epidermis for 3 days (lane 3 figure below) and in primary human keratinocyte culture for 7 days (lane 5 Figure below).

Fig. 6. The experiments show one representative blot (of at least three experiments). The expression level should be better quantified, and compared to the levels from liver cells.

As discussed in the manuscript, thrombin is a serine protease known to be synthesized in the liver only. In order to detect extrahepatic transcription of F2 mRNA, we performed end-point RT-PCR analysis. We fully agree with the reviewer that such analysis does not convey any quantification information, and that was not the purpose of such experiment. The findings clearly revealed a transcription of *F2* mRNA in tissue outside the liver. Finally, we clearly stated in the Figure legend the PCR conditions we used for all gels presented in the Figure 6. Overall, we believe such information in Figure 6 may have a tremendous impact for future research beyond the gastrointestinal tract.

Finally, the language should be improved.

Authors: We have tried to improve the language with the help of native English speakers co-authors of the present paper, who have edited and approved the current revised version of the manuscript.

Reviewer #3 (Remarks to the Author):

General comments

The authors show that epithelial thrombin may dictate host-microbial interactions in the gut and perhaps also in other mucosal tissues. The work is highly novel and may have therapeutic implications. It is recommended that the authors not over-interpret their very nice data. A few additional suggestions for revision are offered.

Authors: We thank the referee for this enthusiastic comment. We have thoroughly revised the text throughout to prevent signs of over-interpretation.

Specific comments

Major

1. To the extent that small numbers of bacteria likely translocate to the liver via the portal circulation in conventionally-raised mice, it is perhaps surprising that hepatic expression of thrombin was unaffected under germ-free conditions. The authors may wish to comment.

Authors: We agree with the reviewer that seeing differential regulation of thrombin expression in the liver and in colon tissues is somehow surprising and we have stated that in the discussion. At present, we have no explanation for this other than a tissue-dependent differential regulation.

As per the unaffected expression of thrombin in the liver of germ-free mice, we have to consider that even germ-free mice are exposed to LPS (contained in food for instance), and therefore are not devoid of bacterial motif stimulation. In germ-free mice, although thrombin mRNA expression is down-regulated, it is not null. One can hypothesize that low exposure to LPS in germ-free mice is sufficient to induce thrombin expression in the liver for hemostatic purposes, but that this low LPS exposure is not sufficient to induce epithelial thrombin expression at the level seen in conventionally-raised mice, just because in the absence of living microbiota, the epithelium does not have to produce large amounts of thrombin for biofilm containment. This is all very speculative and the control of thrombin expression in epithelial cells needs to be investigated in more details. Surprisingly, not very much is known or has been published on the control of thrombin expression in the liver. Studies investigating in parallel the elements controlling thrombin expression in the liver and in intestinal epithelial cells would have to be performed.

In an attempt to elucidate how epithelial thrombin is regulated, we choose to challenge germ-free mice not only with LPS, but also to investigate whether thrombin expression could be recapitulated in germ-free mice recolonized with pathogen-free standard microbiota. We recolonized 5 germfree animals (2 females, 3 males) with cecal content from adult SPF animals by oral gavage, and harvested tissues (liver and colon mucosa) after three weeks. This experiment demonstrated that although GF mice have a dramatic reduction of F2 mRNA transcription in colon mucosa, recolonization with SPF microbiota is sufficient to recover a basal transcription of epithelial thrombin. Epithelial thrombin is thus likely to be regulated by local bacterial metabolites or by bacterial contact. This is in clear contrast with the regulation of F2 mRNA in the liver, which appears to be independent of gut microbial colonization. These results have been added to the new Figure 2 panel D.

Germfree mice (GF, n=5), germfree recolonized with specific-pathogen free (SPF) cecal content (GF>SPF, n=5) and conventionally bred mice (CONV, n=8) were sacrificed at 10-14 weeks old. Relative expression of *F2* mRNA transcript (thrombin) was unchanged in germ-free mouse liver compared to conventionally bred mouse liver and GF>SPF. In germ-free mouse, *F2* mRNA was significantly reduced in colon mucosa compared to conventionally bred mouse colon mucosa. GF recolonized with SPF microbiota had identical transcription of epithelial thrombin compared to conventional mice. One-way ANOVA with Fisher's LSD test ** P < 0.01 versus GF group.

2. Discussion para 5. The authors conclude that thrombin is produced “as a physiological response towards...biofilms”. However, they have not shown this specifically, but rather than thrombin production is reduced in germ-free mice. Since the latter animals have numerous other defects, and also lack other components of the microbiota presumably not involved in biofilm formation, the conclusion is over-stated. The authors are also cautioned against claiming primacy for their findings, particularly since many other epithelial products are produced in response to microbial colonization.

Authors: We agree with the reviewer that this sentence could be perceived as an overstatement. What we meant here was to state that our study shows that in homeostatic physiological conditions (in naïve spf-raised mice and healthy human control tissues), thrombin is produced by intestinal epithelium. We have reworded the sentence in an attempt to be more cautious with our claims.

Minor

1. There are quite a few language and grammatical errors throughout the manuscript, and particularly missing articles, and mixed tenses within a single sentence. The authors are asked to check the manuscript carefully for these issues.

Authors: We have checked the manuscript for language and grammatical errors.

2. The introduction is a single paragraph that stretches for a page and a half. A logical break would be between the introductory material regarding thrombin, and the discussion of the microbiota and biofilms.

Authors: We agree with the reviewer that the introduction could be divided in 2 paragraphs when discussion about microbiota and biofilms start. We have therefore introduced this break.

3. Results para 1 (no page numbers). How was it possible to obtain specifically “human crypts supernatants”?

Authors: We thanks the reviewer for noticing this mistake, it was obviously “human crypts” and not “human crypt supernatants”, we have now corrected this. We have also added page numbers.

4. Results para 2. What is meant by “Thrombin activity was DOSED in supernatants”?

Authors: We thank the reviewer for having noticed this mistake that we have corrected.

5. Results para 3. The information about “threshold cycle” for the liver and colon seems to be misplaced, or at least superfluous.

Authors: We agree with the reviewer that this information does not belong in the result section. This has been corrected accordingly.

6. The figure legends should be expanded to allow the figures to stand alone from the text. For example, in Figure 3, please add the dose of the thrombin inhibitor. All abbreviations should also be defined, and all statistical tests applied must be stated.

Authors: We thank the reviewer for noticing this. We have revised all figure legends accordingly.

7. In every case where a single image is shown, the authors should state the number of times the experiment was repeated with similar findings. This has been done in some cases but not all.

Authors: We fully agree with the reviewer and we have tried to add all these details when they were missing.

8. Results para 5. What is meant by “group’s complex hulls”?

Authors: When plotting the results of the PCoA (principal coordinate analysis) in 2D graphs, it is possible to add ellipsoids and/or convex hulls around the treatment groups of interests. The convex hull represents the areas of the smallest convex polygon containing all points of a specific group (vehicle and lepirudin).

<https://folk.uio.no/ohammer/past/morpho.html>

Graham, R.L.: An Efficient Algorithm for Determining the Convex Hull of a Finite Planar Set. Information Processing Letters (1972) 1, 132-133

Figure 3. Transcriptome analysis of *Muc2*, *Camp*, *Tff3*, *Defb4*, *Reg3g*, *Reg3b*, *Cox2*, *Nos2*, *Tnf*, *Ifng*, *Cxcl1*, *IL17a*, *Adgre1*, *IL1b*, *Cldn1*, *Cldn2*, *Cldn5*, *Tjp1* *Zo1* and *Ocn* genes was performed by qPCR. Principal coordinate analysis with Bray-Curtis dissimilarity matrix illustrates distances between each mouse transcriptome (each dot represents an individual mouse transcriptome), and demonstrate a significant separation between vehicle and lepirudin-treated group (Permanova $P = 0.0379$).

9. Results para 6. It is a bit of a stretch to refer to mesenteric lymph nodes as “distant organs”.

Authors: We totally agree with the reviewer and have corrected this.

REVIEWERS' COMMENTS:

Reviewer #1 (Remarks to the Author):

The authors did a great deal of work to provide new experimental support addressing my comments. With these new data the authors succeeded to overcome my previous concerns.

Reviewer #2 (Remarks to the Author):

The authors have done an impressive job in a relatively short period of time. I particularly appreciate the addition of the requested *in vivo* data on clinically used anticoagulants. It was also nice to see that the comments of all the 3 referees have been addressed very carefully. The paper has been much improved.

Reviewer #3 (Remarks to the Author):

General comments

The authors are to be commended for their thorough revision of their manuscript. The reviewer offers only minor additional suggestions for revision.

Specific comments

1. Supplemental Figure 7. The title of the figure is much too convoluted to be easily understood. In addition, F2 should be defined in the legend. Finally, because the x-axis is on a logarithmic scale, there should be a break in the axis between the values of zero and 0.14 nM.
2. Table 2 and supplemental figure 9. TAILS should be defined. Further, "N-terminomics" seems like jargon – this should be defined more specifically to be appreciated by a broad audience such as the readership of Nature Communications.
3. Supplemental Figure 2. There are numerous grammatical errors in the legend.
4. Grammatical errors persist in the manuscript. Perhaps these could be handled by the copy-editors, but since there are native English speakers amongst the authors, perhaps these errors could again be considered to avoid the possibility that the authors' meaning might be corrupted in the final version.
5. The description of the data in supplemental figure 10 should be moved to the results section, rather than first being mentioned in the discussion.
6. Figure 1. The legend now states that panels B and C are representative of findings in three human donors, whereas Panel C appears to depict data from Caco-2 cells. Please clarify.

Responses to the reviewers

We thank the reviewers for comments and recommendations. We have carefully addressed the issues raised as follows: In this letter, reviewer's comments are in black and our answer appear in red.

Reviewer #3 (Remarks to the Author):

General comments

The authors are to be commended for their thorough revision of their manuscript. The reviewer offers only minor additional suggestions for revision.

We thank the reviewer for this remark.

Specific comments

1. Supplemental Figure 7. The title of the figure is much too convoluted to be easily understood. In addition, F2 should be defined in the legend. Finally, because the x-axis is on a logarithmic scale, there should be a break in the axis between the values of zero and 0.14 nM.

We agree with the reviewer's suggestion. We clarified the title of Supplementary Figure 7, it is now "Reduction of human multispecies microbiota biofilm biomass by thrombin is dependent on its proteolytic activity." In addition, F2 was defined in the legend as Thrombin.

The graph presented in this figure was also modified as requested. A break was introduced on the x axis between 0 and 0.14 nM.

2. Table 2 and supplemental figure 9. TAILS should be defined. Further, "N-terminomics" seems like jargon – this should be defined more specifically to be appreciated by a broad audience such as the readership of Nature Communications.

As requested by the reviewer, TAILS is now defined in the manuscript (TAILS for Terminal amine isotopic labelling of substrates). N-TAILS is a special proteomic approach to detect new protein N-termini compared to a control condition. One bibliographic reference to N-TAILS approaches was already present in the manuscript (Kleifield *et al. Nat Biotechnol.* 2010 Mar;28(3):281-8), and two other references were added in in order for the readers to have access to proper technical details about N-terminomics (Mallia-Milanes *et al. American Journal of Lung and cellular Molecular Physiology*, 2018; 315: 1003-1014. Spitzer, M., Wildenhain, J., Rappsilber, J., and Tyers, M. (2014) BoxPlotR: a web tool for generation of box plots. *Nat. Methods* 11, 121–122).

3. Supplemental Figure 2. There are numerous grammatical errors in the legend.

The legend was corrected.

4. Grammatical errors persist in the manuscript. Perhaps these could be handled by the copy-editors, but since there are native English speakers amongst the authors, perhaps these errors could again be considered to avoid the possibility that the authors' meaning might be corrupted in the final version.

The revised version of the manuscript was reviewed and corrected by native English speakers.

5. The description of the data in supplemental figure 10 should be moved to the results section, rather than first being mentioned in the discussion.

We agree that the first mention of supplementary data (Sup Fig. 10) in the discussion section is not optimal. However, we feel that these data are quite distant from the main message of the manuscript and do not really fit with any part of the result section, or even stand alone as an independent paragraph of the results. They are important for discussion and future work and implications, but would have to be largely completed to stand alone in the result section. Nonetheless, we felt that it was important to mention these data because they were requested originally by one reviewer. Therefore, even if not optimal, we recommend that these data are kept as supplementary data, and are mentioned in the discussion.

6. Figure 1. The legend now states that panels B and C are representative of findings in three human donors, whereas Panel C appears to depict data from Caco-2 cells. Please clarify.

Thank you for the comment, we have now revised the Figure 1 legend according to panel B and C.